# LET MODELS SPEAK CIPHERS:
# MULTIAGENT DEBATE THROUGH EMBEDDINGS

Chau Pham[1*]  Boyi Liu[2*]  Yingxiang Yang[3]  Zhengyu Chen[3]  Tianyi Liu[3]
Jianbo Yuan[3]  Bryan A. Plummer[1†]  Zhaoran Wang[2†]  Hongxia Yang[3†]
[1]Boston University,  [2]Northwestern University,  [3]ByteDance Inc.
{chaupham,bplum}@bu.edu
boyiliu2018@u.northwestern.edu, zhaoranwang@northwestern.edu
{yingxiang.yang,zhengyu.chen,tianyi.liu,jianbo.yuan,hx.yang}@bytedance.com

## ABSTRACT

Discussion and debate among Large Language Models (LLMs) have gained considerable attention due to their potential to enhance the reasoning ability of LLMs. Although natural language is an obvious choice for communication due to LLM's language understanding capability, the token sampling step needed when generating natural language poses a potential risk of information loss, as it uses only one token to represent the model's belief across the entire vocabulary. In this paper, we introduce a communication regime named CIPHER (Communicative Inter-Model Protocol Through Embedding Representation) to address this issue. Specifically, we remove the token sampling step from LLMs and let them communicate their beliefs across the vocabulary through the expectation of the raw transformer output embeddings. Remarkably, by deviating from natural language, CIPHER offers an advantage of encoding a broader spectrum of information without any modification to the model weights, outperforming the state-of-the-art LLM debate methods using natural language by $0.5 - 5.0\%$ across five reasoning tasks and multiple open-source LLMs of varying sizes. This showcases the superiority and robustness of embeddings as an alternative "language" for communication among LLMs. We anticipate that CIPHER will inspire further exploration for the design of interactions within LLM agent systems, offering a new direction that could significantly influence future developments in the field.

## 1 INTRODUCTION

Recent studies in Large Language Models (LLMs) have demonstrated tremendous potential of LLMs to enhance the quality of their responses on reasoning tasks through discussion and debate (Chen et al., 2023; Madaan et al., 2023; Paul et al., 2023; Fu et al., 2023; Jiang et al., 2023; Du et al., 2023; Liang et al., 2023). However, these approaches are often only effective for state-of-the-art LLMs such as GPT-4 (OpenAI, 2023), and have not proven successful with smaller and open-source models such as Vicuna-13B (Chiang et al., 2023). For example, Olausson et al. (2023) found that using self-repair in code generation tasks, is only effective with GPT-4, but proves ineffective when applied to GPT-3.5 (OpenAI, 2022). Similarly, Fu et al. (2023) tested the ability of LLMs in a bargaining game setup and discovered that only a few well-aligned models such as GPT-4 and Claude-v1.3 (Anthropic, 2023) can continuously improve their responses by utilizing feedback. Such inconsistencies in performance across various LLMs motivate our pursuit of a universal solution with consistent efficacy irrespective of the specific LLM employed.

To leverage LLM's language understanding capability, prior work (Du et al., 2023; Liang et al., 2023) adopted natural language during debates. Using natural language for debate is appealing due to its interpretability, but we argue that natural language is neither necessary nor optimal for inter-LLM communication. First, we do not necessarily need to understand the intermediate debates amongst LLMs. Second, natural language generation uses only one token to represent the model's belief over

---

*Equal contribution; work done during an internship at ByteDance.

†Equal advising.

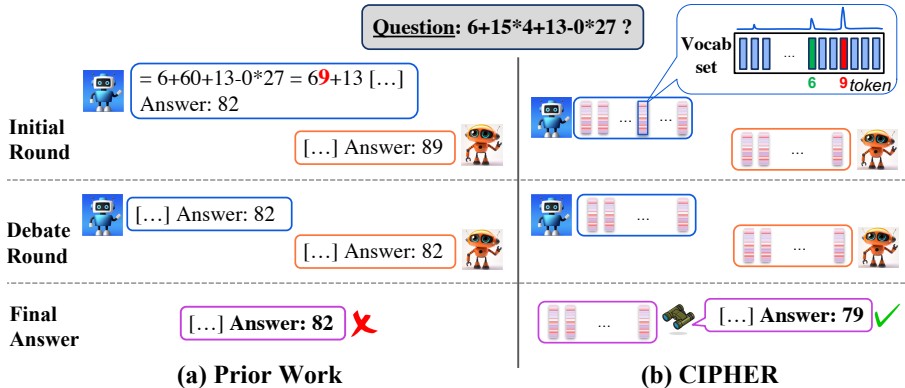

Figure 1: **Comparison of different communication regimes. (a)** LLaMA-2-70B (Touvron et al., 2023b) makes a mistake by generating token "**9**", which should be "**6**" instead. This shows natural language communication between LLMs can lose information. **(b)** In CIPHER (ours), the model outputs embedding vectors and directly receives vectors from other models as inputs. Specifically, instead of sampling one token as seen in prior work, CIPHER generates a vector by taking weighted average of all tokens' embeddings in the vocabulary set. Our vectors provide a richer source of information while also being human interpretable by mapping them back to natural language via a nearest neighbor search over the vocabulary.

the entire vocabulary, which risks losing information embedded within the model output logits. For instance, in reasoning tasks, the most confident token can be wrong. As shown in the top row of Fig. 1a, the model makes a mistake in generating the token "**9**" by picking the most confident token while discarding the valuable information contained in the correct token "**6**".

We address this by proposing a novel communication protocol, Communicative Inter-Model Protocol Through Embedding Representation (CIPHER), that enables LLMs to communicate more freely in the tokenizer's embedding space in the multiagent debate setting. To tackle the issue of information loss in natural language debates, CIPHER lets the models communicate in a vector space, which represents the entire space of potential outputs, thus resulting in a further boost in the performance of debate (Fig. 1b). Specifically, it bypasses the token sampling process by generating the weighted average of all tokens' embeddings in the vocabulary set, instead of yielding a single token. Thus, this passes richer information to other models during debate, particularly when the model is uncertain. While this form of communication deviates from natural language, it still remains interpretable to humans via nearest neighbor search over the vocabulary.

During a debate, LLMs begin by independently providing initial responses to the question (*Initial Round*). Then, each agent receives the other agent's answer to refine its previous answer (*Debate Round*). After some debate rounds, the agents ultimately reach a consensus on a final answer in the majority of cases (Du et al., 2023). We show that LLMs can effectively communicate without relying on natural language and even achieve superior performance.

We validate this claim by evaluating our approach on five diverse datasets across multiple domains: GSM8K (Cobbe et al., 2021), Arithmetic (Du et al., 2023), MMLU Formal Logic, MMLU High School Math, and MMLU Professional Psychology (Hendrycks et al., 2020). Our experiment results show $0.5 - 5.0\%$ performance improvement over the natural-language counterpart. The results align with prior work indicating that when communicating through natural language, weaker models do not outperform majority voting approaches through debates or self-refinement (Wang et al., 2023b). Criticality, in contrast to prior work (Madaan et al., 2023; Du et al., 2023; Liang et al., 2023), we find that our approach can generalize across a wide array of LLMs, enabling even smaller LLMs to unlock the benefits of debate and achieve better performance than majority voting (Wang et al., 2023b). This suggests that, even for open-source models, debate is still an efficient form of communication to boost the performance of LLMs. In summary, our contributions are three-fold:

- We propose CIPHER, a novel inter-model communication protocol for LLMs that share the same tokenizer, regardless of whether they use identical or different embedding-vocabulary mappings.
- We perform comprehensive experiments to validate the efficacy of state-of-the-art debate method through natural language communication (Du et al., 2023) and CIPHER. Our results show that even less powerful LLMs can still benefit from debates.

- We conduct an extensive ablation study to shed light on the mechanisms that make communication through embeddings more effective for debates among LLMs.

## 2 RELATED WORK

**Multiagent debate.** Multiagent debates was pioneered by Du et al. (2023), where LLMs provide initial responses and then make refinements by iteratively considering inputs from peers. Typically, the LLMs reach a consensus in a few debate rounds that is often more accurate. In a concurrent study, Liang et al. (2023) incorporated a "judge" to resolve tiebreakers and determine the final answer. While closely related to the self-improvement via feedback approach where a model iteratively refines its own responses (*e.g.*, Madaan et al. (2023); Akyurek et al. (2023)), this debate approach involves multiple agents with the same role. Thus, it not only encourages divergent thinking in LLMs, but also eliminates the bottleneck at the feedback step, resulting in superior performance compared to self-improvement via feedback methods across diverse datasets. Despite these advances, prior work on multiagent debate have only focused on large and closed-source models such as GPT-4 and GPT-3.5, leaving the efficacy of debate on smaller, open-source models underexplored. To address this gap, our study adapts a multiagent debate setup of Du et al. (2023) and introduce a novel communication protocol in which agents can interact without using natural language.

**Self-improvement via feedback.** Madaan et al. (2023) presented a self-improvement framework, where an LLM doubles as both generator and critic. While this approach allows models such as GPT-3.5 and GPT-4 to boost performance, it falters for smaller and less competent models such as Vicuna-13B (Chiang et al., 2023), which struggled on consistently generating feedback in the required format. Concurrently, Akyurek et al. (2023) introduced RL4F, sidestepping generator weight updates, while Shinn et al. (2023) exploited verbal reinforcement for error correction. Fu et al. (2023) applied self-improvement in a bargaining context, noting a distinct advantage for models like GPT-4 and Claude-v1.3 in iterative improvement. Overall, this body of work highlights the necessity for powerful LLMs as critics and the prevailing limitations in models with parameters fewer than 52B in incorporating natural language feedback effectively (Bai et al., 2022; Saunders et al., 2022).

**Self-debugging for code generation.** Chen et al. (2023); Jiang et al. (2023); Olausson et al. (2023) focused on improving coding ability of LLMs by letting them incorporate explanation of the generated code in natural language and execution results from unit tests to self-debug. Similar to the aforementioned studies, they revealed that models that are more powerful than the generator model yields better results when used as the critic (e.g., GPT-4 gives feedback to GPT-3.5, or humans give feedback to GPT-4). Additionally, Olausson et al. (2023) reported that self-repair on weaker models cannot improve over majority voting (Wang et al., 2023b). In short, these studies underscore a bottleneck in the critic role, necessitating the use of powerful LLMs for generating valuable feedback.

**Reasoning ability in Language Models via prompting.** To further strengthen the reasoning ability of LLMs, Wei et al. (2022) proposed Chain-of-Thought (CoT), a method that employs a series of intermediate reasoning steps to incrementally achieve the final goal. This idea was subsequently generalized by Yao et al. (2023); Long (2023) into Tree-of-Thought, which explores multiple different intermediate steps for the best reasoning path and backtracks when necessary. Graph-of-Thought (Besta et al., 2023) further extends the reasoning ability of LLMs by considering the LLM reasoning as an arbitrary graph, where vertices and edges represent thoughts and their dependencies, respectively. CIPHER incorporates both CoT and few-shot CoT into prompt engineering for boost performance.

## 3 CIPHER: COMMUNICATIVE INTER-MODEL PROTOCOL THROUGH EMBEDDING REPRESENTATION

LLMs function by taking a prompt as input and autoregressively generating a sequence of tokens as the response. From a tokenizer, we have a vocabulary set $\mathcal{V} = \{\texttt{vocab}_i\}_{i \in [V]}$ where $[V] = \{1, \ldots, V\}$ is an index set. Let $\texttt{prompt}$ be the prompt, which contains the instruction, question, and (possible) responses collected in previous debate rounds. We define $\texttt{res}$ to be the generated response from the LLMs. We use $(t)$ in superscripts to indicate the step $t$ of response generation. Correspondingly, we use $(1:t)$ in superscripts to indicate the concatenation of the tokens (or embeddings) generated during the first $t$ steps. A variable is empty whenever it has superscript $(1:t)$ with $t < 1$.

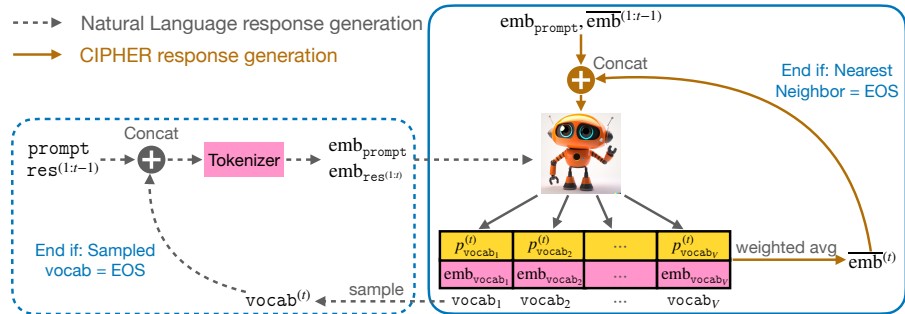

Figure 2: **Response generation in Natural Language and CIPHER**. *Natural Language response generation:* Starting from the top left of the figure, the tokenizer encodes the texts of `prompt` and `res`$^{(1:t-1)}$ into embedding inputs for the LLM. The LLM then outputs a distribution $p^{(t)} = [p^{(t)}_{\text{vocab}_1}, \dots, p^{(t)}_{\text{vocab}_V}]$ over $\mathcal{V}$, from which the next token `vocab`$^{(t)}$ is sampled (bottom mid). *CIPHER response generation:* Instead of sampling a single token `vocab`$^{(t)}$ at step $t$, CIPHER generates a weighted average embedding vector $\overline{\text{emb}}^{(t)}$, utilizing $p^{(t)}$ as weights (Eq. 2). The embedding vector, together with previously generated embeddings $\overline{\text{emb}}^{(1:t-1)}$ and `prompt` embedding $\text{emb}_{\text{prompt}}$, bypasses the tokenizer decoding step and circles directly back into the LLM. Note that CIPHER-generated semantic embeddings approximate a token embedding, but do not precisely match it.

## 3.1 NATURAL LANGUAGE COMMUNICATION

When generating responses, causal LLMs (e.g., LLaMA (Touvron et al., 2023a)) generate tokens one at a time based on the words prior to each token. Given `prompt` and the first $t-1$ generated response tokens `res`$^{(1:t-1)}$, the causal LM calculates $\text{logit}(\texttt{prompt}, \texttt{res}^{(1:t-1)}) \in \mathbb{R}^{1 \times V}$, which is a vector of logits. Then, the next token is sampled from the vocabulary $\mathcal{V}$ with respect to the distribution

$$p^{(t)} = [p^{(t)}_{\text{vocab}_1}, \dots, p^{(t)}_{\text{vocab}_V}] = \text{softmax}\{\text{logit}(\texttt{prompt}, \texttt{res}^{(1:t-1)})/T\}, \qquad (1)$$

where $T > 0$ is the temperature. The distribution $p^{(t)}$ can be viewed as the LLM's belief regarding the most appropriate token at the current position. However, the token sampling step compresses the information of $p^{(t)}$ into a single token, discarding the information on all other tokens in the process. While using one token at each position is useful for humans to understand the outputs from LLMs, we posit that it is not a requirement for effective inter-LLM communication. Recall that in Fig. 1b, we observe the rich information contained in the LLM's belief by the probability distribution over all tokens. Thus, we argue that the token sampling process, which sacrifices information for readability, may lead to a sub-optimal solution for inter-LLM communication. In light of this consideration, we present our CIPHER response generation method for multiagent debate in the section below.

## 3.2 COMMUNICATION THROUGH SEMANTIC EMBEDDINGS

We propose CIPHER, an embedding communication protocol for LLM debates to capture richer encoded information. Our goal is to encode as much information as possible during inter-LLM communication. However, LLMs are designed to understand natural language sequences. Thus, they might not be able to grasp vectors that reside outside the convex hull of the tokenizer's embedding space. To address this, we propose to use the weighted average of embeddings in place of the tokens sampled with respect to $p^{(t)}$ (Eq. 1) in the autoregressive response generation process (Fig. 2).

**CIPHER response generation.** Let $\text{emb}_{\text{prompt}}$ be the embeddings of `prompt`, and $\text{emb}_{\text{vocab}_i} \in \mathbb{R}^{1 \times d}$ be the embedding of token `vocab`$_i \in \mathcal{V}$. Given the concatenation $\overline{\text{emb}}^{(1:t-1)}$ of previously $t-1$ generated semantic embeddings, CIPHER recursively generates new semantic embedding at each step $t = 1, 2, \dots$ by

$$\overline{\text{emb}}^{(t)} = \sum_{i=1}^{V} p^{(t)}_{\text{vocab}_i} \cdot \text{emb}_{\text{vocab}_i}, \qquad (2)$$

$$\text{where} \quad [p^{(t)}_{\text{vocab}_1}, \dots, p^{(t)}_{\text{vocab}_V}] = \text{softmax}\{\text{logit}(\text{emb}_{\text{prompt}}, \overline{\text{emb}}^{(1:t-1)})/T\}. \qquad (3)$$

Here $T > 0$ is the temperature. While the probability $p^{(t)}_{\text{vocab}_i}$ in Eq. 3 bears resemblance to that in Eq. 1, CIPHER bypasses the tokenizer and directly passes semantic embeddings into LLMs instead of

---

**Algorithm 1** CIPHER Debate

---

1: **Input:** Question and instructions `prompt`, number of rounds $R \geq 2$, and $n$ CIPHER debaters $\{D_i\}_{i \in [n]}$.

——————————————————————— (*initial round*) ———————————————————————

2: Obtain embedding of the prompt, $\text{emb}_{\text{prompt}}$, via the tokenizer
  **For debater** $i = 1, \ldots, n$**:**
   3: Get initial CIPHER response $\text{cipher}_i \leftarrow D_i(\text{emb}_{\text{prompt}})$ from debater $i$ by (2)
  **EndFor**

——————————————————————— (*debate rounds*) ———————————————————————

  **For round** $r = 2, \ldots, R$**:**
   4: Get updated prompt embedding $\text{emb}_{\text{prompt}} \leftarrow \text{concat}(\text{emb}_{\text{prompt}}, \text{cipher}_1, \ldots, \text{cipher}_n)$
    **For debater** $i = 1, \ldots, n$**:**
     5: Get CIPHER response $\text{cipher}_i \leftarrow D_i(\text{emb}_{\text{prompt}})$ from debater $i$ by (2)
    **EndFor**
  **EndFor**

——————————————————————— (*post processing*) ———————————————————————

6: **Output:** Final response $\text{cipher}^* = $ *Convert-and-Aggregate*$(\text{cipher}_1, \ldots, \text{cipher}_n)$

---

taking raw texts as inputs. The generation process stops either (i) the EOS token embedding becomes the nearest neighbor of the newly generated embedding, or (ii) the maximal sequence length is reached. For a CIPHER response generation process that stops after $\tau$ steps, the CIPHER response is defined as $\text{cipher} = \overline{\text{emb}}^{(1:\tau)}$, which is the concatenation of all the generated semantic embeddings.

**CIPHER debate.** We formalize the CIPHER debate procedure in Algorithm 1. First, we convert the question and instructions into embeddings $\text{emb}_{\text{prompt}}$ using the tokenizer (Line 2). Then, for each debate round, we form an embedding representation by concatenating $\text{emb}_{\text{prompt}}$ and (possible) CIPHER responses, $\text{cipher}_i$, from all debaters in previous rounds (Line 4). This embedding representation is then input directly into the models without the token decoding step. The debaters then generate refined CIPHER responses following Eq. 2 (Line 5). To close the debate, at *Convert-and-Aggregate* step (Line 6), we convert the embedding responses back to natural language using nearest neighbor search over the vocabulary set, then aggregate them to obtain the final response. In most cases, LLM debaters typically reach a consensus answer by the final round, as observed in Du et al. (2023). When divergence in final responses occurs, majority voting (Wang et al., 2023b) or random tie-breaking are often used. However, majority voting may not be suitable for open-ended questions (*e.g.*, summarization tasks) where multiple correct answers exist, as in *the game of 24* (Yao et al., 2023), and scenarios where debates involving only two agents. Thus, in our experiments, we select the response from the debater with the lowest temperature as the final answer. This approach achieves comparable accuracy to the best performing debater while maintaining computational efficiency by running inference on only one model in the final round, as shown in Fig. 5.

**Role of temperature.** The temperature $T$ in Eq. 1 and Eq. 3 controls the smoothness of the probability $p_{\text{vocab}_i}^{(t)}$. When $T \rightarrow 0$, both CIPHER's embedding generation and natural language generation result in greedy generation. In contrast, a large $T$ leads to a uniform averaging and sampling over the whole vocabulary set for CIPHER and natural language generation, respectively. Choosing proper temperatures for the debaters plays a pivotal role in the performance of CIPHER and natural language debate. Thus, to ensure fairness of our empirical evaluation, we utilize Bayesian optimization (Nogueira, 2014) to select the best performing temperatures for each method in our experiments in Section 4. Moreover, we conduct sensitivity analysis on the temperatures in Section 5.2.

**Intuition.** The idea behind CIPHER is connected to Expected SARSA (Sutton & Barto, 1998) in reinforcement learning. Specifically, Expected SARSA replaces the sampled Q-values in vanilla SARSA (Rummery & Niranjan, 1994; Sutton, 1995) with the expected values over all next actions, leading to significant advantages over vanilla SARSA (Van Seijen et al., 2009). For causal LLMs, the autoregressive token generation process can be viewed as a Markov decision process where, at the step $t$, the state is the previously generated response $\text{res}^{(1:t-1)}$, the action space is the vocabulary set $\mathcal{V}$, the policy is $p^{(t)}$ defined in Eq. 1, and the reward is tied to the response's accuracy. To this end, our weighted averaging of embeddings shares the same spirit as Expected SARSA, computing expectations over possible tokens. Meanwhile, the natural language response generation process, where tokens are probabilistically sampled, aligns with vanilla SARSA.

## 4 EXPERIMENTS

### 4.1 EXPERIMENTAL SETUP

**Baseline methods.** We benchmark our proposed approach against the following three baselines:

- **Single Answer**: a single LLM provides one response to the given question in natural language.
- **Self-Consistency** (Wang et al., 2023b): a single LLM independently generates multiple responses to the given question, then applies majority voting to determine the final answer.
- **Natural Language Debate (NLD)** (Du et al., 2023): each LLM first provides an initial response to the given question. Subsequently, the LLMs use each other's responses to refine their previous responses (see Appendix A for the formal algorithm). Note this approach serves as the most direct baseline for CIPHER (ours), differing primarily in terms of the communication protocol.

Due to the difficulties experienced by open-source models in generating appropriately formatted feedback as a critic, as discussed in Section 2, we do not include self-improvement via feedback methods (*e.g.*, Madaan et al. (2023)) in the baselines. For all the methods, we use Bayesian optimization to select temperatures, which are reported in Appendix D for reproducibility of our results.

**Models.** We conduct most of our experiments using LLaMA2-70B (Touvron et al., 2023b), as it is one of the largest open-source models with an extended context window of up to 4,096 tokens. Additionally, to test our approach's robustness and generalizability, we conduct experiments with various other models, including LLaMA-65B (Touvron et al., 2023a), Falcon-40B-Instruct (Penedo et al., 2023), MPT-30B (Team, 2023), and WizardMath-70B-V1.0 (Luo et al., 2023; Xu et al., 2023).

**Datasets.** We evaluate CIPHER Debate on five reasoning datasets that span across four different domains. (i) **GSM8K** (Cobbe et al., 2021) consists of a variety of grade school math problems created by human problem writers. (ii) MMLU (Hendrycks et al., 2020) we pick three datasets from three different categories, **Formal Logic** dataset from the Humanities category, **High School Math** dataset from the STEM category, and **Professional Psychology** dataset from the Social Science category. (iii) **Arithmetic**: following Du et al. (2023), we evaluate mathematical expressions comprising six unique two-digit numbers that include addition, multiplication, and subtraction operations. For large datasets (GSM8K, Professional Psychology, and Arithmetic), we tune the temperature on a validation set of 200 sampled questions and evaluate on another 200 questions in a separate test set.

**Prompts for initial and debate rounds.** We combine few-shot examples with chain-of-thought prompting (Wei et al., 2022) and zero-shot instruction ("Let's think step by step") (Kojima et al., 2022) to encourage agents to generate both the final answer and the reasoning steps. This boosts response accuracy and provides valuable information for other LLMs during debates. Additionally, CIPHER is compatible with various prompting methods. See Appendix E for detailed prompts.

**Metrics.** For debates among identical LLMs with different temperatures, we measure the accuracy of the final answer. To evaluate the efficacy of debates among different LLMs that share the same tokenizer, we show more detailed results by testing the correctness of all the final-round answers.

### 4.2 RESULTS

Following Du et al. (2023), we assess the performance of CIPHER debates against the baselines in 3-round debates between 2 LLMs. Below we provide a detailed discussion of our results.

**Comprehensive evaluation with the LLaMA family.** We comprehensively evaluate CIPHER debate using the LLaMA family of LLMs (LLaMA-65B (Touvron et al., 2023a) and LLaMA2-70B (Touvron et al., 2023b)) across five reasoning datasets. Table 1 presents the results from debates between two identical LLaMA family LLMs operating at different temperatures. Both Self-Consistency (Major@5) and NLD (Du et al., 2023) exhibit significant performance enhancements over a single answer. Remarkably, CIPHER consistently outperforms both baselines, achieving a $1.0 - 5.0\%$ boost over NLD across all datasets. We evaluate debates based on the final responses of the agent with a lower temperature, resulting in five responses per debate. For fair comparisons, our self-consistency baselines (labeled as *Major@5*) also use five responses. Additionally, while all the baseline methods display high variance ($0.5 - 3.0\%$ across datasets) due to their token sampling process, CIPHER's deterministic embedding generation ensures consistent outputs.

Table 1: **Debate accuracies (%) between two identical LLaMA family models at different temperatures with 3 rounds.** Except *Single Answer* baseline, each baseline generates 5 responses per question. Both Self-Consistency (*Major@5*) and *NLD* improve the performance over the *Single Answer* baseline. CIPHER further widens the gap, outperforming NLD by $1.0 - 5.0\%$ consistently.

| Model | Method | GSM8K | H.S. Math | Psychology | Formal Logic | Arithmetic |
|---|---|---|---|---|---|---|
| LLaMA2-70B | Single Answer | 60.0±2.3 | 38.3±2.6 | 73.6±1.2 | 46.0±2.9 | 79.5±0.3 |
| | Major@5 (Wang et al., 2023b) | 64.3±1.4 | 41.3±1.5 | 74.0±0.7 | 44.4±2.3 | 79.7±0.3 |
| | NLD (Du et al., 2023) | 64.8±2.4 | 39.4±0.9 | 74.2±0.7 | 49.2±0.9 | 81.1±0.8 |
| | CIPHER (ours) | **66.0**±0.0 | **41.5**±0.0 | **75.0**±0.0 | **52.4**±0.0 | **85.0**±0.0 |
| LLaMA-65B | Single Answer | 50.8±1.6 | 33.8±1.8 | 68.8±1.5 | 43.5±2.7 | 27.6±1.1 |
| | Major@5 (Wang et al., 2023b) | 52.7±3.3 | 36.7±0.7 | 70.5±0.4 | 46.8±2.1 | 29.8±0.9 |
| | NLD (Du et al., 2023) | 51.7±1.4 | 36.7±0.9 | 70.0±2.0 | 46.0±1.7 | 30.4±0.4 |
| | CIPHER (ours) | **52.9**±0.0 | **38.5**±0.0 | **70.9**±0.0 | **50.8**±0.0 | **33.0**±0.0 |

Table 2: **Debate accuracies (%) between LLaMA2-70B and LLaMA-65B on (a) Arithmetic and (b) GSM8K** datasets. *Agreement* indicates the consensus (%) of the two debaters (i.e., they generate the same response). CIPHER debates result in higher accuracies in both debaters compared to NLD.

| **(a) Arithmetic** | NLD (Du et al., 2023) | | | CIPHER (ours) | | |
|---|---|---|---|---|---|---|
| | LLaMA2-70B | LLaMA-65B | Agreement | LLaMA2-70B | LLaMA-65B | Agreement |
| Round 1 | 73.0 | 32.5 | 31.5 | 73.5 | 35.0 | 34.5 |
| Round 2 | 69.0 | 56.0 | 61.5 | 70.0 | 61.5 | 69.0 |
| Round 3 | 72.0 | 60.5 | 77.5 | **74.5** | **62.5** | 78.0 |

| **(b) GSM8K** | NLD (Du et al., 2023) | | | CIPHER (ours) | | |
|---|---|---|---|---|---|---|
| | LLaMA2-70B | LLaMA-65B | Agreement | LLaMA2-70B | LLaMA-65B | Agreement |
| Round 1 | 61.0 | 51.3 | 51.5 | 61.5 | 48.0 | 44.0 |
| Round 2 | 58.8 | 57.5 | 77.5 | 64.3 | 60.3 | 70.3 |
| Round 3 | 61.8 | 59.0 | 88.8 | **64.8** | **63.3** | 83.8 |

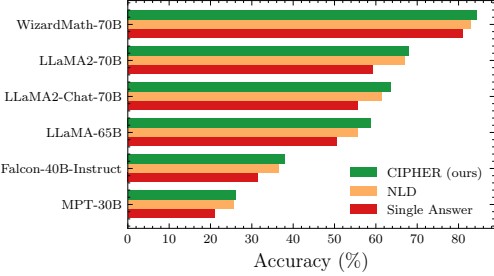

Figure 3: **Multiagent debates across different models on GSM8K.** Debate in Natural Language (NLD) (Du et al., 2023) improves the performance of various open-source models compared to the direct generation of a single answer. CIPHER provides an additional performance boost, ranging from 0.5% to 3.5% across different models.

Table 2 presents the results of debates between LLaMA-65B and LLaMA2-70B on Arithmetic and GSM8K datasets. While the tokenizer is shared between LLaMA-65B and LLaMA2-70B, they use distinct embeddings for each vocabulary. To tackle this issue, we keep a mapping vocab → $[\text{emb}_{\texttt{LLaMA1}}, \text{emb}_{\texttt{LLaMA2}}]$ for each vocabulary and compute the weighted average of the embeddings using the embeddings of the receiver. For example, to pass the message to LLaMA2-70B, we average over $\text{emb}_{\texttt{LLaMA2}}$ during the CIPHER response generation from LLaMA-65B debater. This guarantees that the output of LLaMA-65B is encoded within the LLaMA2-70B's token embedding space. While debating proves beneficial for both agents, LLaMA-65B experiences a more substantial improvement, especially on Arithmetic dataset, with an increase from 35% to 62.5% (Table 2(a), CIPHER).

**Different LLMs on selected datasets.** To substantiate the efficacy of CIPHER further, we conduct additional debates using three other open-source LLMs: Falcon-40B-Instruct (Penedo et al., 2023), MPT-30B (Team, 2023), and WizardMath-70B-V1.0 (Luo et al., 2023; Xu et al., 2023). The experiments are performed on GSM8K dataset, as shown in Fig. 3. NLD (Du et al., 2023) enhances the performance of a wide range of open-source models compared to generating a single answer. CIPHER further provides additional performance boosts ranging from 0.5% to 3.5% across various models, which are significant considering there is no modification to the model weights.

## 5 ANALYSIS AND DISCUSSION

In this section, we provide experimental results of debates in extended scales, debate temperature sensitivity analysis, and an ablation study through partial CIPHER implementation. Additional

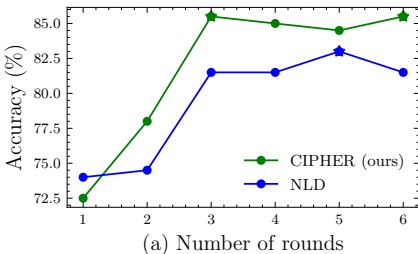 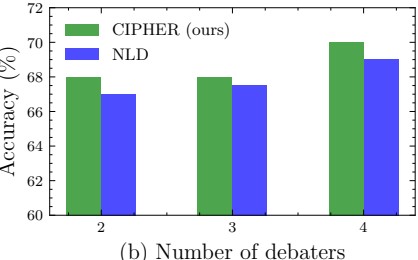

Figure 4: **(a) Number of rounds.** In general, adding more rounds helps to boost performance on both CIPHER and NLD (Du et al., 2023). Stars (★) indicate the best performance of 2 LLaMA2-70B debaters on Arithmetic dataset. **(b) Number of debaters.** Adding more debaters is beneficial for both CIPHER and NLD (Du et al., 2023). We report the accuracies of LLaMA2-70B on GSM8K dataset.

experimental results on positional bias and performance bounds of debates are in Appendix B. We also present qualitative analysis of CIPHER debates on a few selected questions in Appendix C.

## 5.1 DEBATE IN EXTENDED SCALES

While recent work has shown the phenomenon of rapidly diminishing marginal return of scaling up the numbers of debaters and rounds (Du et al., 2023; Liang et al., 2023), we nonetheless include these scenarios for the sake of a comprehensive evaluation. This allows us to paint a more complete picture of the performance landscape of CIPHER debate. Fig. 4a shows that adding more debate rounds can eke out a small gain for debate methods at the cost of more responses. Likewise, from Fig. 4b, we observe that involving more debaters is beneficial to the debate with a similar trade-off.

## 5.2 TEMPERATURE SENSITIVITY

In this section, we delve into the performance dynamics of debates regarding temperature selection (Fig. 5). In particular, we investigate the advantages of allowing certain debaters to deviate from natural language during debates. More concretely, one debater operates at a lower temperature to ensure the final answer remains comprehensible to humans, while the other debater is tasked with conveying information that may deviate from natural language by generating its responses at a higher temperature. We employ Bayesian optimization (Nogueira, 2014) to identify promising pairs of temperatures in debates between two LLaMA2-70B debaters on Arithmetic dataset. Unlike the final response aggregation strategy in CIPHER, we evaluate the effectiveness of these temperature pairs based on the accuracy of the final response from the first debater (*temperature 1*), which can operate at a higher temperature. Such an experiment setting not only sheds light on the impact of temperature selection on debate performance but also guides our final response aggregation strategy.

In general, the debate is more beneficial when the debaters are more diverse. For NLD (Du et al., 2023) (Fig. 5, top row), optimal performance is often achieved with all the temperatures set below 1. This can be attributed to the inherent information loss during the token sampling process in natural language generation. At higher temperatures, LLMs are more likely to produce nonsensical responses, potentially weakening stronger debaters' performance. In contrast, for CIPHER, optimal performance is often obtained at wider spread apart temperatures, as shown in Fig. 5a, bottom row. Notably, the optimal regions for CIPHER are mostly on the left side of the charts, indicating that CIPHER benefits the most when it pairs a low-temperature agent with a high-temperature one. At higher temperatures, the probability distribution of the tokens in the vocabulary becomes more uniform, allowing CIPHER's responses to lean towards less confident token choices. This effectively complements the information communicated by the other lower-temperature debater, which focuses on more confident tokens. Additionally, low-temperature agents are necessary for collecting results that can be interpretable by humans. Therefore, a good strategy for CIPHER is to employ various temperature agents and use the response of the lower temperature agent as the final debate answer.

## 5.3 ABLATION STUDY ON CIPHER

To unveil the mechanisms contributing to the performance gain brought by CIPHER, we invoke CIPHER response generation only at the positions where the model exhibits high uncertainty $U^{(t)}$ regarding the next token. Setting a threshold $\varepsilon > 0$, we adopt CIPHER generation whenever $U^{(t)} > \varepsilon$

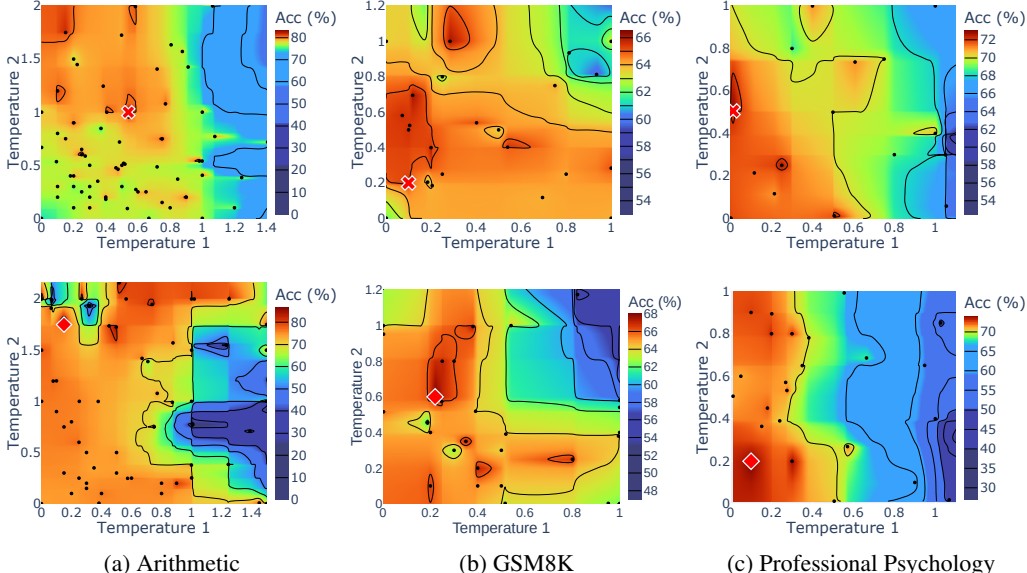

(a) Arithmetic         (b) GSM8K         (c) Professional Psychology

Figure 5: **2D-contour plots of accuracy over different temperatures.** We depict the performance debates between two LLaMA2-70B CIPHER debaters across different pairs of temperatures, where cross marks (✖) and diamonds (♦) denote the best performance of NLD (Du et al., 2023) (top row) and CIPHER (ours, bottom row), respectively. We report the debate performance based on the final responses generated by debater 1 (*temperature 1*). For CIPHER (bottom row), the optimal regions appear on the left side of the charts, where *temperature 1* is lower than *temperature 2*. These indicate that a good strategy is using various temperature agents, and choosing the response of the lower temperature agent as the final debate answer. See Section 5.2 for more discussion.

Figure 6: **Partial CIPHER.** We invoke CIPHER for token generation in some steps while opting for greedy sampling in others, based on generation uncertainty. Experiments with LLAMA2-70B pairs on the Arithmetic dataset.

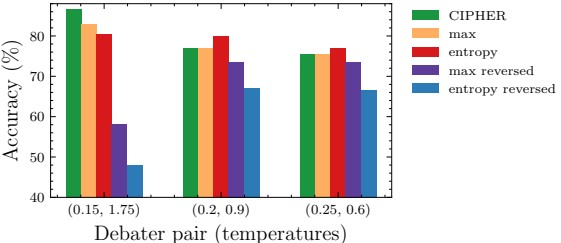

and default to greedy sampling otherwise. To quantify the uncertainty $U^{(t)}$ at a given position $t$, we employ two variants (i) *entropy* $U^{(t)} = -\sum_{i=1}^{V} p_i^{(t)} \log p_i^{(t)}$ of the generation distribution $p^{(t)}$ over the vocabulary set, or (ii) *max* probability $U^{(t)} = 1 - \max_{i \in [V]} p_i^{(t)}$. Additionally, we explore two reversed variants, *entropy reversed* and *max reversed*, which invoke CIPHER when $U^{(t)} \le \varepsilon$. Fig. 6 shows that partial application of CIPHER in *max* and *entropy* aligned closely with full CIPHER application, endorsing our hypothesis that CIPHER advantages from the information retention during the moments of uncertainty. In contrast, *entropy reversed* and *max reversed* see huge performance drops, showing that diverging from CIPHER can drastically diminish the efficacy of debates.

# 6 CONCLUSION

Our proposed CIPHER demonstrates promising results across various reasoning tasks. Without necessitating special training, CIPHER enhances the debate efficacy of a wide range of LLMs, outperforming majority voting and debate methods using natural language. Our study highlights the potential of LLMs' belief information and serves as the first step towards unlocking this potential in LLM debates, closing the gap between proprietary models and open-source models. It remains intriguing whether there is an even more efficient way of transferring belief information among LLMs, given the fact that LLMs are only trained to intake the embeddings of natural language text tokens.

## LIMITATIONS AND BROADER IMPACTS

One limitation of CIPHER lies in its applicability, which is currently restricted to Language Models (LLMs) sharing a common vocabulary set. Expanding our approach to encompass LLMs with distinct tokenizers would require meticulous alignment of their respective embedding-vocabulary mappings. Such alignment is challenging, as tokenizers often employ significantly different text segmentation strategies. For instance, while one tokenizer may break a word into subwords or characters, another might treat it as a single token. We recognize this as a promising avenue for future research. Overcoming this limitation could pave the way for constructing even more robust and efficient LLM agent systems, potentially unlocking unprecedented collaborative capabilities among diverse LLMs. On a positive note, the open-source LLM community is seeing rapid growth of families of open-source LLMs with shared tokenizers such as LLaMA family (LLaMA, LLaMA2, LLaMA2-chat, and Code LLaMA). As such, we are optimistic that our work, in its current form, will continue to contribute to the growth and evolution of the community.

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

## A    MULTIAGENT DEBATE FRAMEWORK

We demonstrate the procedure of natural language debate in Algorithm 2. We use $\|$ to denote concatenation. Here, the debaters $\{D_i\}_{i \in [n]}$ are conventional LLMs and the responses $\texttt{res}_i$ are in natural language. In contrast, the debaters $\{D_i\}_{i \in [n]}$ in Algorithm 1 are CIPHER debaters and the responses $\text{emb}\texttt{res}_i$ are embeddings generated by (2). We also note that, since the responses generated in Algorithm 2 are all in natural language, we can directly aggregate (denoted as $\texttt{Aggregate}$) the final round responses without performing a nearest neighbor search over the vocabulary set.

---

**Algorithm 2** Multiagent Natural Language Debate

---

    **Input:** Question and instructions $\texttt{prompt}$, number of rounds $R \geq 2$, and $n$ LLM debaters $\{D_i\}_{i \in [n]}$.
    **For debater** $i = 1, 2, \ldots, n$**:**
        Get initial natural language response $\texttt{res}_i \leftarrow D_i(\texttt{prompt})$ from debater $i$.
    **EndFor**
    **For round** $r = 1, \ldots, R$**:**
        Get updated prompt $\texttt{prompt} \leftarrow \text{concat}(\texttt{prompt}, \texttt{res}_1, \ldots, \texttt{res}_n)$.
        **For debater** $i = 1, \ldots, n$**:**
            Get updated natural language response $\texttt{res}_i \leftarrow D_i(\texttt{prompt})$.
        **EndFor**
    **EndFor**
    $\texttt{res}^{\star} \leftarrow \texttt{Aggregate}(\texttt{res}_1, \ldots, \texttt{res}_n)$
    **Output:** Final response $\texttt{res}^{\star}$

---

## B    EXTRA EXPERIMENTS

### B.1    INVESTIGATING POSITIONAL BIAS IN DEBATES

The issue of positional bias in utilizing LLMs as evaluators has attracted increasing attention in recent studies (Wang et al., 2023a; Zheng et al., 2023; Liusie et al., 2023). Although our multiagent debate setting differs from "LLMs as evaluators," we recognize that the sequence in which prior rounds responses are fed into subsequent rounds of debates could still have non-negligible effects on the outcomes of the debates. Recall that in debate rounds, for CIPHER, we feed the responses of other debaters first, then the response of the debater itself. In Figure 7, we provide a further investigation into the positional bias within our multiagent debate setting by swapping the order, *i.e.*, a variant of CIPHER where other debaters' responses are fed after its own response. We find that the effect of positional bias is negligible when the two debaters operate at similar temperatures. However, when the debaters are dissimilar, swapping the order of responses can result in a significant difference. Both NLD Du et al. (2023) and CIPHER show better performance when the debaters are more diverse.

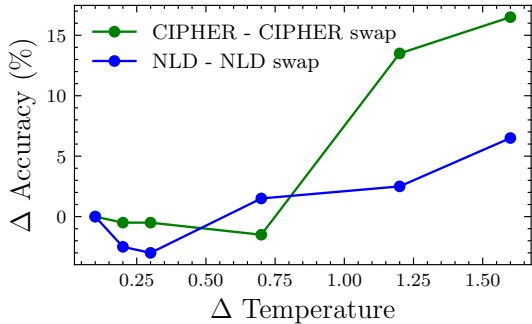

Figure 7: **Effect of the order of responses in two-agent debate.** The green line shows the difference between CIPHER and its variant when swapping the order of responses (CIPHER swap), while the blue line shows the difference between NLD (Du et al., 2023) and its variant (NLD swap).

## B.2 PERFORMANCE BOUNDS OF DEBATES

Prior work showcases significant performance enhancements achieved through LLM debates. Yet, the limit of such a performance gain remains a captivating facet of study. To investigate the performance upper bound that can be achieved through multiagent debate, we conduct an experiment where the LLM debater is pitted against an expert debater. We proxy the expert debater by having it always give the ground truth answers. Conversely, to establish the performance lower bound, we conduct an experiment where the LLM debater constantly receives nonsensical feedback from other debaters. Specifically, we employ 2 dummy debaters, one with extremely high temperature that gives non-sense, and the other gives non-relevant responses by using misaligned ground truth answers from other questions in the batch. Figure 8 illustrates these bounds on GSM8K with LLaMA2-70B and its much less capable version, LLaMA2-7B.

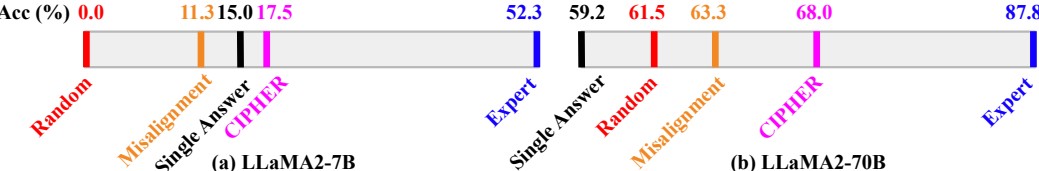

Figure 8: **Upper bound and lower bound of two-agent CIPHER debate on GSM8K.** A debater generates with **Random** (Extremely high temperature debater), **Misalignment** (debater responds misaligned ground-truth), **Single Answer**, **CIPHER** (ours), and **Expert** (debater responds ground-truth) on GSM8K dataset. We observe that debate can be detrimental when the model has low capacity (Fig. *a*), but it does not pose much harm in the case of a more powerful model (Fig. *b*).

## B.3 ATTENTION HEATMAPS

Figure 9 shows a comparison of attention heatmaps for CIPHER and NLD (Du et al., 2023) at the $45^{th}$ decoder layer of LLaMA2-70B. These heatmaps correspond to the arithmetic question we used in Figure 1 during the last debate round. Specifically, we compute the similarity between the $q$ vector of the last token of the first agent and the $k$ vectors of its preceding 100 tokens. We can observe that NLD's heatmap exhibits uniform attention distribution, lacking intense focus on any particular segment. Conversely, CIPHER's heatmap shows some distinct bright spots, particularly around the $40^{th}$ attention head and the $74^{th}$ time step. This suggests that the model's attention is highly focused on those areas, potentially indicating areas of higher relevance for the task.

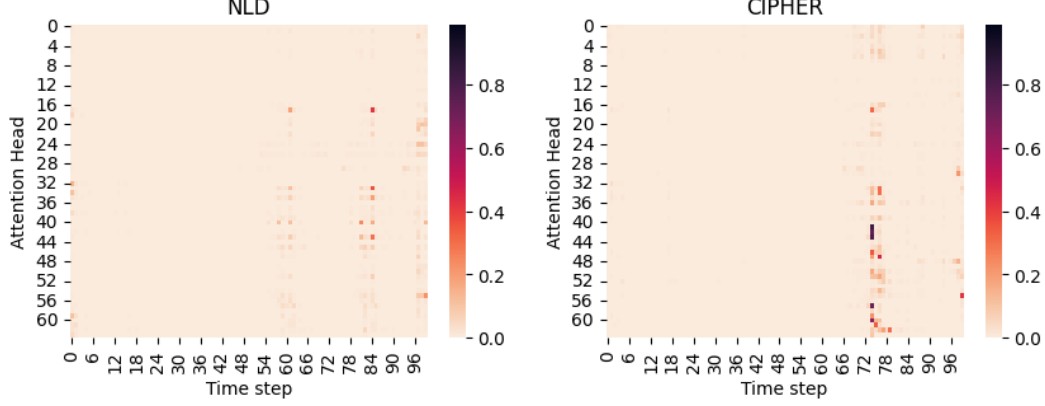

Figure 9: **Attention heatmaps of CIPHER and NLD (Du et al., 2023) when using 2 LLaMA2-70B.** The heatmap generated by CIPHER indicates some significant bright spots, especially around the $40^{th}$ attention head and the $74^{th}$ time steps.

## C  QUALITATIVE RESULTS

We present some detailed results of each debater's response within the context of a two-agent debate of NLD (Du et al., 2023) and our method, CIPHER. Figure 10 and Figure 11 display the complete example used to demonstrate the content shown in Figure 1 in the main paper. In Figure 12, we present a debate utilizing CIPHER on the GSM8K dataset.

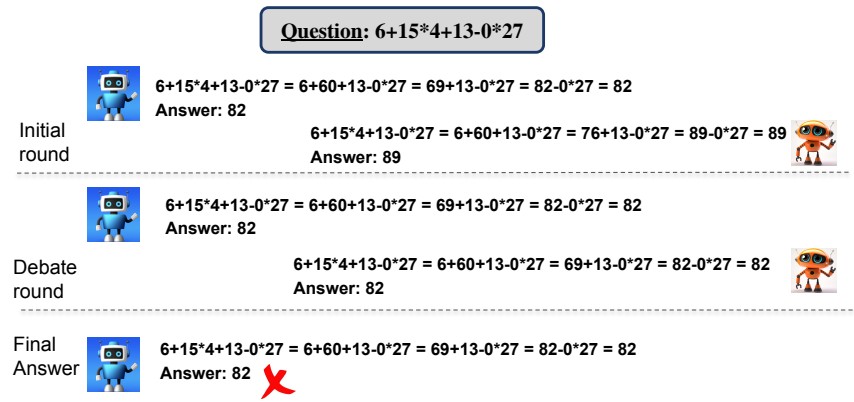

Figure 10: **Example of NLD on Arithmetic dataset.** A complete debate involving two LLaMA2-70B debaters using a temperature pair of (0.00, 0.67), as presented in Figure 1.

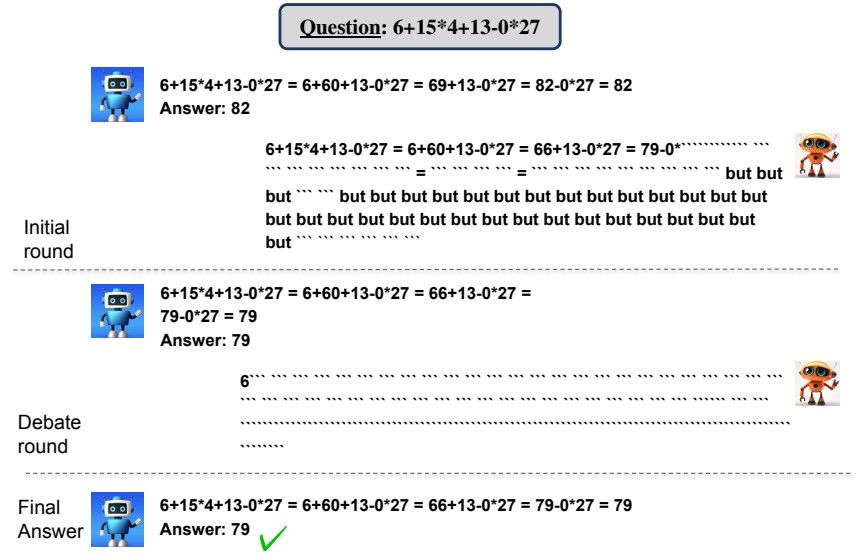

Figure 11: **Example of CIPHER on Arithmetic dataset.** A complete debate involving two LLaMA2-70B debaters using a temperature pair of (0.25, 1.75), as presented in Figure 1. We convert the generated embeddings back to natural language using a nearest neighbor search on the vocabulary.

**Question:** A tank has a capacity of 18000 gallons. Wanda and Ms. B decided to pump water from a pond to fill the tank in two days. On the first day, working in shifts, Wanda filled 1/4 of the tank's capacity with water, and Ms. B pumped 3/4 as much water as Wanda pumped into the tank that day. On the second day, Wanda pumped 2/3 of the amount of water she pumped on the previous day, while Ms. B only pumped 1/3 of the number of gallons she pumped on the first day. How many gallons of water are remaining for the tank to be full?

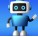 On the first day, Wanda pumped 18000*1/4=4500 gallons of water into and Ms. B pumped 18000*3/4=13500 gallons of water. On the second day, Wanda pumped 18000*2/3=12000 gallons of water, and Ms. B pumped 18000*1/3=6000 gallons of water. In total, Wanda pumped 4500+12000=16500 gallons of water, and Ms. B pumped 13500+6000=19500 gallons of water. So total, 16500+19500=36000 gallons of water were pumped. Since the tank has a capacity of 18000 gallons, there18000-36000=-0 gallons of water are remaining for the tank to be full.
**Answer: 0**

**Initial round**

On the first day, Wanda pumped 18000*1/4=4500 gallons of water into and Ms. B pumped 38000*3/4=4500 gallons of water. On 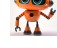 the second day, Wanda pumped 18000*2/3=12000 gallons of water, and Ms. B pumped 18000*1/3=6000 gallons of water. In total, they they they they they they they they they they they they they they they they they they they they they they they they they they they they they they they they they they they they they they they they they they they they they they they they they they they they they they they they they they they they they they they

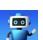 On the first day, Wanda pumped 18000*1/4=4500 gallons of water, and Ms. B pumped 18000*3/4=13500 gallons of water. On the second day, Wanda pumped 18000*2/3=12000 gallons of water, and Ms. B pumped 18000*1/3=6000 gallons of water. In total, Wanda pumped 4500+12000=16500 gallons of water, and Ms. B pumped 13500+6000=19500 gallons of water. In total, 16500+19500=36000 gallons of water were pumped. Since the tank has a capacity of 18000 gallons, 18000-36000=--18000 gallons of water are remaining for the tank to be full.
**Answer: -18000**

**Debate round**

On the first day, Wanda pumped 1/4 of the tank's capacity with and and and and and and and and and and and and and and and and and and and and and and and and and and and and and and and and and and and and and and and and and and and and and and and and and and and and and and and and and and and and and and and and and and and and and and and and and and and and and and and and and and and and and and and and and and and and and and and and and and and and and and and and and and and and and and and and and and and and and and and and and and and and and and and and and and and and and and and and and and and and and and and and and and and and and and and and and and and and and and

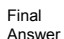 On the first day, Wanda pumped 1/4 of the tank's capacity with water, and Ms. B pumped 3/4 as much water as Wanda pumped into the tank that day. So, Wanda pumped 1/4 of 18000 gallons = 4500 gallons of water, and Ms. B pumped 3/4 of 4500 gallons = 3375 gallons of water. On the second day, Wanda pumped 2/3 of the amount of water she pumped on the previous day, while Ms. B only pumped 1/3 of the number of gallons she pumped on the first day. So, Wanda pumped 2/3 of 4500 gallons = 3000 gallons of water, and Ms. B pumped 1/3 of 3375 gallons = 1125 gallons of water. In total, Wanda pumped 4500 + 3000 = 7500 gallons of water, and Ms. B pumped 3375 + 1125 = 4500 gallons of water. In total, 7500 + 4500 = 12000 gallons of water were pumped. Since the tank has a capacity of 18000 gallons, 18000 - 12000 = 6000 gallons of water are remaining for the tank to be full. Answer: 6000

**Final Answer** ✅

Figure 12: **Example of CIPHER on GSM8K dataset.** A complete debate involving two LLaMA2-70B debaters using a temperature pair of (0.40, 1.00).

# D   DETAILED EXPERIMENT SETUPS

## D.1   TEMPERATURE

To ensure the reproducibility of our results, we include the temperatures of the debaters in each of our experiments in the following.

**Temperatures in the comprehensive evaluation with LLaMA family LLMs.** Corresponding to Table 1, Table 3 presents the temperatures of LLaMA family debaters during the debates.

Table 3: Temperatures of two identical LLaMA family models during debates.

| Model | Method | GSM8K | High School Math | Psychology | Formal Logic | Arithmetic |
|---|---|---|---|---|---|---|
| LLaMA2-70B | Single Answer | 0.15 | 0.24 | 0.33 | 0.30 | 0.35 |
| | Major@5 Wang et al. (2023b) | 0.80 | 0.30 | 0.30 | 0.50 | 0.60 |
| | NLD Du et al. (2023) | (0.10, 0.20) | (0.30, 0.49) | (0.01, 0.51) | (0.10, 0.20) | (0.54, 1.00) |
| | CIPHER (ours) | (0.22, 0.60) | (0.10, 0.82) | (0.10, 0.20) | (0.10, 0.20) | (0.15, 1.75) |
| LLaMA-65B | Single Answer | 0.20 | 0.30 | 0.10 | 0.20 | 0.40 |
| | Major@5 Wang et al. (2023b) | 0.80 | 0.80 | 0.80 | 0.60 | 0.80 |
| | NLD Du et al. (2023) | (0.10, 0.20) | (0.40, 0.50) | (0.30, 0.40) | (0.20, 0.85) | (0.08, 0.20) |
| | CIPHER (ours) | (0.25, 0.85) | (0.49, 0.80) | (0.10, 0.40) | (0.25, 0.85) | (0.67, 1.43) |

Corresponding to Table 2, Table 4 presents the temperatures of LLaMA-65B and LLaMA2-70B during the debates on the Arithmetic and GSM8K datasets.

Table 4: Temperatures of LLaMA2-70B and LLaMA-65B during the dabtes on **(a) Arithmetic** and **(b) GSM8K** datasets.

| (a) Arithmetic | NLD Du et al. (2023) | | CIPHER (ours) | |
|---|---|---|---|---|
| | LLaMA2-70B | LLaMA-65B | LLaMA2-70B | LLaMA-65B |
| Temperature | 0.32 | 0.51 | 0.35 | 0.35 |

| (b) GSM8K | NLD Du et al. (2023) | | CIPHER (ours) | |
|---|---|---|---|---|
| | LLaMA2-70B | LLaMA-65B | LLaMA2-70B | LLaMA-65B |
| Temperature | 0.20 | 0.40 | 0.35 | 0.63 |

**Temperatures in evaluation across different LLMs.** Corresponding to Figure 3, Table 5 presents the temperatures of open-source LLMs on GSM8K dataset.

Table 5: Temperatures of 2-agent debate across different models on GSM8K

| Method | LLaMA2-70B | LLaMA2-Chat-70B | LLaMA-65B | Falcon-40B-Instruct | MPT-30B | WizardMath-70B |
|---|---|---|---|---|---|---|
| Single Answer | 0.15 | 0.15 | 0.20 | 0.40 | 0.45 | 0.00 |
| NLD (Du et al., 2023) | (0.10, 0.20) | (0.20, 0.40) | (0.10, 0.20) | (0.20, 0.40) | (0.35, 0.62) | (0.15, 0.35) |
| CIPHER (ours) | (0.22, 0.60) | (0.25, 0.65) | (0.25, 0.85) | (0.25, 0.65) | (0.23, 0.64) | (0.26, 0.69) |

**Temperatures in debates in extended scales.** Corresponding to Figure 4, we report the temperatures of LLaMA2-70B debaters in the debates in extended scales in Table 6 and Table 7.

Table 6: Temperatures of LLaMA2-70B debaters in debates in extended scales on GSM8K dataset in reported Figure 4(a).

| Method | 2 debaters |
|---|---|
| NLD (Du et al., 2023) | (0.013, 1.072) |
| CIPHER | (0.498, 1.725) |

Table 7: Temperatures of LLaMA2-70B debaters in debates in extended scales on GSM8K dataset reported in Figure 4(b).

| Method | 2 debaters | 3 debaters | 4 debaters |
|---|---|---|---|
| NLD (Du et al., 2023) | (0.100, 0.500) | (0.300, 0.500, 0.700) | (0.442, 0.176, 0.745, 0.539) |
| CIPHER | (0.250, 0.600) | (0.001, 0.725, 1.067) | (0.641, 0.464, 0.507, 0.202) |

## D.2 COMPUTATION RESOURCE

For LLaMA family debates, we use $4\times$ NVIDIA A100 SXM 80GB GPUs as the major computation resource.

## E DEBATE PROMPTS

In this section, we provide the detailed prompts we use for each dataset. The prompts consist of two parts:

- Initial round prompt: it prompts the debaters to generate their initial response with a chain-of-thought explanation.
- debate round prompt: it prompts debaters to incorporate the responses generated in the previous rounds and provide refined responses.

## E.1 GSM8K

### E.1.1 INITIAL ROUND PROMPT

We employ the following 3-shot prompt for all models, with the exception of WizardMath-70B-V1.0 (Luo et al., 2023; Xu et al., 2023), for which we use the CoT prompt provided by the authors [1].

**LLaMA, Falcon, MPT**

```
Can you solve the math question? Explain your reasoning step by step.
Write your final answer as a numerical number, in the form '''Answer:
{number}''', at the end of your response.

Question: '''A robe takes 2 bolts of blue fiber and half that much white
fiber.  How many bolts in total does it take?'''
Solution is '''It takes 2/2=1 bolt of white fiber. So the total amount of
fabric is 2+1=3 bolts of fabric.
Answer: 3'''

---
Question: '''James decides to run 3 sprints 3 times a week.  He runs 60
meters each sprint. How many total meters does he run a week?'''
Solution is '''He sprints 3*3=9 times. So he runs 9*60=540 meters.
Answer: 540'''

---
Question: '''Steve loves playing video games.  His parents get him a
console along with 5 games for his birthday.  He saves up enough money to
buy 1 game per month for a year, and then the following year he starts
buying 2 games a month.  For the third year he buys 4 games a month as
he has a new part-time job that makes him more money.  He also gets 5
games for Christmas every year.  How many games does Steve have after 3
years?'''
Solution is '''Steve buys 1 game a month his first year, and since there
are 12 months in a year that means he buys 12*1=12 games. Steve then
buys 2 games per month for a year, and since there are 12 months in a
year that means he buys 12*2=24 games. Steve then buys 4 games per month
for a year, and since there are 12 months in a year that means he buys
12*4=48 games. Steve also gets 5 games for Christmas each year for three
years, for another 5*3=15 games. In total over three years, Steve gets
12+24+48+15=99 games. Since Steve started with 5 games, he has 5+99=104
games after 3 years.
Answer: 104'''

---
Question: '''{QUESTION_PLACEHOLDER}'''
Solution is '''
```

**WizardMath**

```
Below is an instruction that describes a task. Write a response that
appropriately completes the request.

### Instruction:
{QUESTION_PLACEHOLDER}

### Response: Let's think step by step.
```

---

[1]https://github.com/nlpxucan/WizardLM/tree/main/WizardMath

E.1.2 DEBATE ROUND PROMPT

We utilize the same 3-shot prompt below for all models, except for WizardMath-70B-V1.0, where we use a zero-shot prompt according to the authors' instructions.

**LLaMA, Falcon, MPT**

```
Using the solution from other agent as additional information, update
your solution to the question to have a correct answer.

Question: '''James decides to run 3 sprints 3 times a week. He runs 60
meters each sprint. How many total meters does he run a week?'''

Agent 1's solution: '''James runs 3*60*3=1800 meters a week.
Answer: 1800'''

Your solution: '''James runs 3 sprints, each 60 meters, 3 times a week.
So he runs a total of 3*60*3=540 meters in a week.
Answer: 540'''

Correct solution: '''Let's think step by step:
James runs 3 sprints per session, and he does this 3 times a week. So, in
one week, he runs a total of 3 * 3 = 9 sprints.
Since each sprint is 60 meters long, James runs a total of 9 * 60 = 540
meters.
Answer: 540'''
---

Question: '''Ali is a baker. Leila ordered 3 chocolate cakes for $12 each
and 6 strawberry cakes for $22 each.  How much should Leila pay Ali?'''

Agent 1's solution: '''Ali should charge Leila $12*3=$36 for the
chocolate cakes and $22*6=$132 for the strawberry cakes.
Therefore, Leila should pay Ali $36+$132=$168.
Answer: 168'''

Your solution: '''The cost of 3 chocolate cakes at $12 each is: 3 * 12 =
36
The cost of 6 strawberry cakes at $22 each is: 6 * 22 = 132
So, the total cost of the cakes is 132.
Answer: 132'''

Correct solution: '''Let's think step by step:
The cost of 3 chocolate cakes at $12 each is: 3 * 12 = $36
The cost of 6 strawberry cakes at $22 each is: 6 * 22 = $132
Total amount is: $36 + $132 = $168
Therefore, Leila should pay Ali $168.
Answer: 168'''
---

Question: '''Jeff and Brad are trying to divide 100 dollars between them.
Jeff gets 4 times as much as Brad. How much does Jeff get in dollars?'''

Agent 1's solution: '''According to the information provided, Jeff gets 4
times as much as Brad. So, Jeff would get 4x dollars.
Since the total amount they have is 100 dollars, we can set up an
equation:
x + 4x = 100
Combine like terms:
5x = 100
Now, solve for x:
x = 100 / 5
x = 20
So, Brad gets 20 dollars. Now, to find how much Jeff gets:
Jeff's amount = 4 * Brad's amount = 4 * 20 = 20
```

```
So, Jeff gets 20 dollars.
Answer: 20'''

Your solution: '''Let x be the amount of money Brad gets.
Then Jeff gets 4x. We know that the total amount of money is 100 dollars,
so we can set up the equation 4x + x = 100.
Solving for x, we get x = 25. Therefore, Brad gets 25 dollars and Jeff
gets 4*25=100 dollars
Answer: 100'''

Correct solution: '''Let's think step by step:
Let's assume that Brad gets x dollars.
Jeff gets 4 times as much as Brad. So, Jeff would get 4x dollars.
We can set up an equation: x + 4x = 100
Combine like terms: 5x = 100
Now, solve for x: x = 100 / 5 = 20
So, Brad gets 20 dollars. Now, to find how much Jeff gets:
Jeff's amount = 4 * Brad's amount = 4 * 20 = 80
So, Jeff gets 80 dollars.
Answer: 80'''
---

Question: '''{QUESTION_PLACEHOLDER}'''

Agent 1's solution: '''{OTHER_SOLUTION_PLACEHOLDER_0}'''

Your solution: '''{MY_SOLUTION_PLACEHOLDER}'''

Correct solution: '''Let's think step by step:
```

**WizardMath**

```
Below is an instruction that describes a task. Write a response that
appropriately completes the request.

### Agent 1's solution:
{OTHER_SOLUTION_PLACEHOLDER_0}

### Your solution:
{MY_SOLUTION_PLACEHOLDER}

### Instruction:
Use the responses above, answer the following math question.
{QUESTION_PLACEHOLDER}

### Response: Let's think step by step.
```

## E.2   MMLU - HIGH SCHOOL MATH

### E.2.1   INITIAL ROUND PROMPT

```
Can you solve the question below? Explain your reasoning step by step.
Write a letter as your final answer, in the form '''Answer: {letter}''',
at the end of your response.

Question: '''When a spaceship full of scientists landed on Planet Q, they
found that $\frac{17}{40}$ of the $160$ aliens had $3$ eyes. How many
aliens had $3$ eyes?
A. 67
B. 35
C. 36
D. 68'''
```

Solution: ```Let's think step by step:
The fraction of aliens with 3 eyes is given as 17/40.
We can calculate the number of aliens with 3 eyes as follows:
Number of aliens with 3 eyes = (17/40) * 160
Simplifying this expression, we get:
Number of aliens with 3 eyes = (17 * 160) / 40 = 2720 / 40 = 68
Therefore, the number of aliens on Planet Q who had 3 eyes is 68, closest
to answer choice D.
Answer: D```
---

Question: ```If f : (x, y)  (x + y, 2y  x) for every coordinate pair in
the xy-plane, for what points (x, y) is it true that f : (x, y)  (x, y)?
A. (0, 0) only
B. The set of points (x, y) such that x = 0
C. The set of points (x, y) such that y = 0
D. The set of points (x, y) such that x = y```

Solution: ```We have that f(x, y) = (x + y, 2y  x).
The only way for this to equal (x, y) is if x + y = x and 2y  x = y.
The first equation simplifies to y = 0, and the second equation
simplifies to x = 0.
Therefore, the only point that works is (0,0), closest to answer choice
A.
Answer: A```
---

Question: ```Alex grows an initial culture of 100 Rhizopus stolonifer
fungi on a sample of bread.
She wants to model the growth of the fungi according to the exponential
equation A = Pe^(rt), where A is the final number of fungi, P is the
initial number, r is the growth rate, and t is time elapsed in hours.
If after 5 hours she measures the number of fungi to be 750, what is the
value of r?
A. 0.403
B. 0.863
C. 2.015
D. 4.317```

Solution: ```We are given that the initial number of fungi is P = 100 and
after 5 hours, the number of fungi is A = 750.
Substituting these values into the equation, we have:
750 = 100e^(5r)
To solve for r, we can divide both sides of the equation by 100:
7.5 = e^(5r)
Next, we take the natural logarithm (ln) of both sides:
ln(7.5) = ln(e^(5r))
Using the property of logarithms, we can bring down the exponent:
ln(7.5) = 5r
Now, we divide both sides by 5 to isolate r:
r = ln(7.5) / 5
We find that r = 0.403
Therefore, the value of r is approximately 0.403, closest to answer
choice A.
Answer: A```
---

Question: ```{QUESTION_PLACEHOLDER}```

Solution: ```

E.2.2  DEBATE ROUND PROMPT

Using the solution from other agent as additional information, update
your solution to the question to have a correct answer.

Question: ```If f : (x, y)  (x + y, 2y  x) for every coordinate pair in
the xy-plane, for what points (x, y) is it true that f : (x, y)  (x, y)?
A. (0, 0) only
B. The set of points (x, y) such that x = 0
C. The set of points (x, y) such that y = 0
D. The set of points (x, y) such that x = y```

Agent 1's solution: ```Given that f : (x, y) -> (x + y, 2y  x), we can
equate the components of the two tuples:
x + y = x, and
2y - x = y.
Simplifying these equations, we find:
y = 0, and
y = x.
From the equation y = 0, we can conclude that for any value of x, the
point (x, 0) will satisfy f, which is closest to answer choice C.
Answer: C```

Your solution: ```The function f maps a point (x, y) to the point (x +
y, 2y  x). For f to map a point to itself, we need x + y = x and 2y  x
= y. Solving the first equation, we get x = 0. Substituting x = 0 into
the second equation, we get y = 0. Therefore, the only point that maps to
itself is (0, 0). So the answer is
(0,0) only, which is closest to answer choice A.
Answer: A```

Correct solution: ```Let's think step by step:
We have that f(x, y) = (x + y, 2y  x).
The only way for this to equal (x, y) is if x + y = x and 2y  x = y.
The first equation simplifies to y = 0, and the second equation
simplifies to x = 0.
Therefore, the only point that works is (0,0), closest to answer choice
A.
Answer: A```
---

Question: ```Let $h(4x-1) = 2x + 7$. For what value of $x$ is $h(x) = x$?
A. 420
B. 69
C. 7
D. 15```

Agent 1's solution: ```We have that h(4x-1) = 2x + 7.
Let a=4x-1, then x=(a+1)/4.
Substituting a=4x-1 into the first equation, we get
h(a)=2((a+1)/4)+7=(a+1)/2+7=a/2+7.5
We want to find h(a)=a, so we have an equation a/2+7.5=a.
Solving for a, we get a/2=7.5. This gives a=7.5*2=15, which is closest to
answer choice D.
Answer D```

Your solution: ```We are given that h(4x1)=2x+7. We want to find the
value of x such that h(x)=x.
Substituting x for 4x1 in the first equation, we get h(x)=2(x1)+7=x+5.
Solving x+5=x, we get x=5, which is closest to answer choice C.
Answer C```

Correct solution: ```Let's think step by step:
We have the equation $h(4x-1) = 2x + 7$.
Let's define $a = 4x-1$. This implies $x = \frac{a+1}{4}$.

```
Substituting $a$ into the equation, we have $h(a) =
2\left(\frac{a+1}{4}\right) + 7 = \frac{a}{2} + 7.5$.
Substituting $x$ into $a$, we have  $h(x) =  \frac{x}{2} + 7.5$.
We want $h(x) = x$, thus we can set up the equation $\frac{x}{2} + 7.5 =
x$.
Simplifying this equation, we can subtract $\frac{x}{2}$ from both sides:
$7.5 = \frac{x}{2}$.
To solve for $x$, we multiply both sides by $2$:
$2 \times 7.5 = x$.
This gives us $x = 15$, which is closest to answer choice D.
Answer: D'''
---

Question: '''{QUESTION_PLACEHOLDER}'''

Agent 1's solution: '''{OTHER_SOLUTION_PLACEHOLDER_0}'''

Your solution: '''{MY_SOLUTION_PLACEHOLDER}'''

Correct solution: '''Let's think step by step:
```

### E.3   MMLU - PROFESSIONAL PSYCHOLOGY

#### E.3.1   INITIAL ROUND PROMPT

```
Question: '''Which of the following statements best exemplifies
criterion-referenced (as opposed to norm-referenced) measurement
A. Alice answered 63% of the items correctly
B. Susans score was average for her class
C. James ranked at the 86th percentile on the Scholastic Aptitude Test
(SAT)
D. Joe received a z score of 1.6'''

Solution: '''Criterion-referenced measurement focuses on assessing an
individual's performance based on predetermined criteria or standards. In
this statement, Alice's performance is evaluated based on the percentage
of items she answered correctly, which directly relates to the criteria
of correctness. The criterion for success is set at 63%, and Alice's
performance is measured against that criterion.
Therefore, the answer is A.
Answer: A'''
---

Question: '''Group A and Group B each are composed of equal numbers of
men and women. Both groups read a prose passage. Group A is told that it
was written by a man, and Group B is told that it was written by a woman.
Both groups then rate the quality of the prose. What are the likely
results
A. The males in Group B will rate the prose as significantly poorer than
will the males in Group A
B. The females in Group B will rate the prose as significantly poorer
than will the females in Group A
C. Group A will rate the prose as significantly better than will Group B
D. The females in Group A will rate the prose 2s significantly poorer
than will the females in Group B; the males in Group A will rare the
prose as significantly better than will the males in Group B'''

Solution: '''The experiment involves telling each group of participants a
different author's gender for the same prose passage. In this case, Group
A is told that the passage was written by a man, while Group B is told it
was written by a woman. The likely result is that Group A, who believes
the passage was written by a man, will rate the prose as better compared
to Group B, who believes it was written by a woman. This indicates a
```

potential bias or preconceived notions about the quality of writing based
on the author's gender.
Therefore, the answer is C.
Answer: C'''
---

Question: '''Dr. Blitz has been seeing a client for four months when the
client says that he is very dissatisfied with the progress of therapy. To
be consistent with the provisions of the ethics codes published by the
American and Canadian Psychological Association, Dr. Blitz should:
A. tell the client that he can't expect to make major changes in his life
in only four months.
B. encourage the client to schedule more frequent sessions so they can
work more intensely on his problems.
C. help the client work through his resistance.
D. discuss the matter with the client and terminate therapy and provide
him with referrals if he desires.'''

Solution: '''When a client expresses dissatisfaction with the progress
of therapy, it is important for the psychologist to address the issue
ethically. The appropriate course of action is to have an open and
honest discussion with the client about their concerns. Dr. Blitz should
explore the client's dissatisfaction, understand their perspective, and
discuss potential options for addressing their needs. This may include
terminating therapy and providing referrals to other professionals
who may be better suited to meet the client's needs or addressing the
concerns in a way that maintains the therapeutic relationship. Ultimately,
the decision should be made collaboratively between Dr. Blitz and the
client based on the client's best interests.
Therefore, the answer is D.
Answer: D'''
---

Question: '''The keyword method is most useful for:
A. remembering word pairs.
B. remembering a long list of unrelated words.
C. recalling the order of mathematical operations.
D. mastering spatial relationships.'''

Solution: '''The keyword method is a mnemonic strategy commonly used
to remember word pairs or foreign vocabulary. It involves associating
a familiar word or image (the keyword) with the target word or concept
to be remembered. By creating a vivid mental connection between the
keyword and the target word, it becomes easier to recall the information
later. This method is particularly effective for memorizing word pairs or
connecting new words with their meanings.
Therefore, the answer is A.
Answer: A'''
---

Question: '''{QUESTION_PLACEHOLDER}'''

Solution: '''

### E.3.2 DEBATE ROUND PROMPT

Using the solution from other agent as additional information, update
your solution to the question to have a correct answer.

Question: '''Group A and Group B each are composed of equal numbers of
men and women. Both groups read a prose passage. Group A is told that it
was written by a man, and Group B is told that it was written by a woman.

Both groups then rate the quality of the prose. What are the likely
results
A. The males in Group B will rate the prose as significantly poorer than
will the males in Group A
B. The females in Group B will rate the prose as significantly poorer
than will the females in Group A
C. Group A will rate the prose as significantly better than will Group B
D. The females in Group A will rate the prose 2s significantly poorer
than will the females in Group B; the males in Group A will rare the
prose as significantly better than will the males in Group B```

Agent 1's solution: ```The experiment involves telling each group of
participants a different author's gender for the same prose passage. In
this case, Group A is told that the passage was written by a man, while
Group B is told it was written by a woman. The likely result is that
Group A, who believes the passage was written by a man, will rate the
prose as better compared to Group B, who believes it was written by a
woman. This indicates a potential bias or preconceived notions about the
quality of writing based on the author's gender.
Therefore, the answer is C.
Answer: C```

Your solution: ```There is a phenomenon called gender attribution bias.
Gender attribution bias is the tendency to judge the quality of a piece
of writing based on the gender of the author. In this case, the males in
Group A will be more likely to rate the prose as good if they are told
that it was written by a man, and the females in Group B will be more
likely to rate the prose as bad if they are told that it was written by a
woman.
Therefore, the answer is D.
Answer: D```

Correct solution: ```Let's think carefully.
Assume that individuals may have preconceived notions or biases regarding
the quality of writing based on the author's gender. In this case, Group
A, which believes the passage was written by a man, is likely to rate the
prose as better compared to Group B, which believes it was written by
a woman. These results suggest a potential gender bias influencing the
perceived quality of the prose. It is important to note that individual
differences and other contextual factors can also play a role in shaping
the outcomes of such experiments.
Therefore, the answer is C.
Answer: C```
---

Question: ```The effects of crowding are influenced by a number of
factors, including all of the following except ___________.
A. Self-esteem
B. Age
C. Gender
D. Culture```

Agent 1's solution: ```Self-esteem is not typically considered a direct
factor that influences the effects of crowding. While self-esteem can
have an impact on an individual's perception and reaction to various
situations, it is not specifically identified as a primary factor that
directly influences the effects of crowding. The other options, including
age, gender, and culture, are commonly recognized as factors that can
influence how individuals perceive and respond to crowded environments.
Therefore, the answer is A.
Answer: A```

Your solution: ```The effects of crowding can be influenced by factors
such as gender, culture, and self-esteem. However, age is not typically
considered a significant factor that directly influences the effects

of crowding. While people of different age groups may have varying
preferences or reactions to crowded environments, age itself is not a
primary determinant of how individuals experience or respond to crowding.
Therefore, the answer is B.
Answer: B```

Correct solution: ```Let's think carefully.
The effects of crowding are influenced by a number of factors, including
all of the following except Age.
Age is not typically considered a significant factor that directly
influences the effects of crowding. While individuals of different ages
may have varying preferences or reactions to crowded environments, age is
not generally regarded as a primary factor that consistently determines
the impact of crowding. Factors such as self-esteem, gender, and culture,
on the other hand, are more commonly acknowledged as influences on how
individuals perceive and respond to crowded situations.
Therefore, the answer is B.
Answer: B```
---

Question: ```{QUESTION_PLACEHOLDER}```

Agent 1's solution: ```{OTHER_SOLUTION_PLACEHOLDER_0}```

Your solution: ```{MY_SOLUTION_PLACEHOLDER}```

Correct solution: ```Let's think step by step:

## E.4  MMLU - FORMAL LOGIC

### E.4.1  INITIAL ROUND PROMPT

Can you solve the question below? Explain your reasoning step by step.
Write a letter as your final answer, in the form ```Answer: {letter}```,
at the end of your response.

Question: ```Which of the given formulas of PL is the best symbolization
of the following sentence?
Elliott likes to skateboard or he likes to write songs and to play them.
A. S  (W  P)
B. (S  W)  P
C. S  (W  P)
D. (S  W)  P```

Solution: ```The sentence is saying that Elliott likes to skateboard
or he likes to write songs and play them. This is symbolized by a
disjunction (), which means "or."
The first disjunct is simply S, which states that Elliott likes to
skateboard. The second disjunct is (W  P), which is the conjunction of W
and P. This means that Elliott likes to write songs and play them
The best symbolization of the sentence is A. S  (W  P).
Thus, the correct answer choice is A.
Answer: A```
---

Question: ```Use indirect truth tables to determine whether each set of
propositions is consistent. If the set is consistent, choose an option
with a consistent valuation. (There may be other consistent valuations.)
(A  B)  C
˜C
˜A Ł B
A. Inconsistent
B. Consistent. Consistent valuation when A and B are true and C is false

C. Consistent. Consistent valuation when A and C are true and B is false
D. Consistent. Consistent valuation when B and C are true and A is
false```

Solution: ```We can construct the truth table for this set of
propositions as follows:
A | B | C | (A  B)  C | ˜C | ˜A Ł B |
---|---|---|---|---|---|
T | T | F | T | T | T |
T | T | T | T | F | T |
T | F | F | T | T | T |
F | T | F | T | T | T |
F | F | F | T | T | T |
F | F | T | T | F | T |
F | F | T | F | T | F |
The last column of the truth table shows that the set of propositions is
consistent.
Thus, the correct answer choice is B.
Answer: B```
---

Question: ```Select the best English interpretation of the given
proposition, using the following translation key: Ax: x is an apartment
Hx: x is a house Lx: x is large Bxy: x is bigger than y (x){(Lx  Ax) Ł
(y)[(Hy  ˜Ly)  Bxy]}
A. All large apartments are bigger than all houses that are not large.
B. Some house that is not large is bigger than all large apartments.
C. Any apartment bigger than a house that is not large is large.
D. Every large apartment is bigger than some house that is not large.```

Solution: ```We have:
Ax: x is an apartment
Hx: x is a house
Lx: x is large
Bxy: x is bigger than y
We have translation of the proposition:
x: For all x.
(Lx  Ax) Ł: If x is large and x is an apartment, then
(y)[(Hy  ˜Ly)  Bxy]: there exists a y such that y is a house, y is not
large, and x is bigger than y.
This means that for all apartments that are large, there exists a house
that is not large that the apartment is bigger than.
Thus, the correct answer choice is D.
Answer: D```
---

Question: ```{QUESTION_PLACEHOLDER}```

Solution: ```

### E.4.2 DEBATE ROUND PROMPT

Using the solution from other agent as additional information, update
your solution to the question to have a correct answer.

Question: ```Which of the given formulas of PL is the best symbolization
of the following sentence?
Elliott likes to skateboard or he likes to write songs and to play them.
A. S  (W  P)
B. (S  W)  P
C. S  (W  P)
D. (S  W)  P```

Agent 1's solution: '''The best symbolization of the given sentence
"Elliott likes to skateboard or he likes to write songs and to play them"
is option B: (S ∨ W) ∧ P.
Thus, the correct answer is B.
Answer: B'''

Your solution: '''The sentence is saying that Elliott likes to skateboard
or he likes to write songs and play them. In other words, he likes to do
one of those things, but not necessarily both.
Option A correctly captures this meaning. It says that Elliott likes to
skateboard or he likes to write songs and play them. This is the only
option that does not imply that Elliott likes to do both things.
Thus, the correct answer is A.
Answer A'''

Correct solution: '''Let's think step by step:
The sentence is saying that Elliott likes to skateboard or he likes to
write songs and play them. This is symbolized by a disjunction (∨), which
means "or."
The first disjunct is simply S, which states that Elliott likes to
skateboard. The second disjunct is (W ∧ P), which is the conjunction of W
and P. This means that Elliott likes to write songs and play them
The best symbolization of the sentence is A. S ∨ (W ∧ P).
Thus, the correct answer is A.
Answer: A'''
---

Question: '''Use indirect truth tables to determine whether each set of
propositions is consistent. If the set is consistent, choose an option
with a consistent valuation. (There may be other consistent valuations.)
(G ∨ H) Ł H
˜H ⊃ I
G ⊃ ˜I
A. Inconsistent
B. Consistent. Consistent valuation when G, H, and I are true
C. Consistent. Consistent valuation when G and H are true and I is false
D. Consistent. Consistent valuation when G is true and H and I are
false'''

Agent 1's solution: '''We can use the following truth table to determine
the consistency of the set of propositions.
We can see that the set of propositions is consistent, because there is
at least one line in the truth table where all of the propositions are
true. The consistent valuation is shown in the row where G, H, and I are
all false.
Thus, the correct answer is D.
Answer: D'''

Your solution: '''Option C is correct because there is a consistent
valuation where G and H are both true and I is false. The consistent
valuation is G = T, H = T, and I = F.
Thus, the correct answer is C.
Answer: C'''

Correct solution: '''Let's think step by step:
We have the indirect truth table for the given set of propositions:
G | H | ˜H | I | (G ∨ H) | (G ∨ H) Ł H | ˜H ⊃ I | G ⊃ ˜I | ˜H ⊃ I / G ⊃ ˜I
-- | -- | -- | -- | -- | -- | -- | -- | -- | --
T | T | F | F | T | T | F | F | T | F
T | T | F | T | T | T | F | F | T | T
T | F | T | F | F | F | T | F | T | T
T | F | T | T | F | F | T | F | T | T
F | T | F | F | T | T | F | T | F | T
F | T | F | T | T | T | F | T | F | T
F | F | T | F | T | T | F | F | F | F

```
F | F | T | T | F | F | T | F | F | F
The set of propositions is consistent if and only if there is at least
one row in the truth table where all the propositions are true. As you
can see, there is no row where all the propositions are true. Therefore,
the set of propositions is inconsistent.
Option D is correct because the propositions are inconsistent, so there
is no consistent valuation where G is true and H and I are both false.
Thus, the correct answer is D.
Answer: D'''
---

Question: '''{QUESTION_PLACEHOLDER}'''

Agent 1's solution: '''{OTHER_SOLUTION_PLACEHOLDER_0}'''

Your solution: '''{MY_SOLUTION_PLACEHOLDER}'''

Correct solution: '''Let's think step by step:
```

## E.5 ARITHMETIC

### E.5.1 INITIAL ROUND PROMPT

```
Calculate the result of the expression step by step. Write your final
answer as a numerical number, in the form '''Answer: {number}''', at the
end of your response.

Question: '''What is the result of 14+22*27+6-13*18?'''
Solution: '''14+22*27+6-13*18 = 14+594+6-234 = 608+6-234 = 614-234 = 380
Answer: 380'''
---

Question: '''What is the result of 23+25*6+23-1*28?'''
Solution: '''23+25*6+23-1*28 = 23+150+23-28 = 173+23-28 = 196-28 = 168
Answer: 168'''
---

Question: '''What is the result of 5+19*12+27-15*28?'''
Solution: '''5+19*12+27-15*28 = 5+228+27-420 = 233+27-420 = 260-420 =
-160
Answer: -160'''
---

Question: '''{QUESTION_PLACEHOLDER}'''
Solution: '''
```

### E.5.2 DEBATE ROUND PROMPT

```
Using the solution from other agent as additional information, update
your solution to the question to have a correct answer.

Question: '''What is the result of 14+22*27+6-13*18?'''

Agent 1's solution: '''14+22*27+6-13*18 = 14+594+6-234 = 608+6-234 =
614-234 = 300
Answer: 300'''

Your solution: '''14+22*27+6-13*18 = 14+594+6-234 = 608+6-234 = 614-234 =
380'
Answer: 380'''

Correct solution: '''Let's think step by step:
```

```
14+22*27+6-13*18 = 14+594+6-234 = 608+6-234 = 614-234 = 380
Answer: 380'''
---

Question: '''What is the result of 23+25*6+23-1*28?'''

Agent 1's solution: '''23+25*6+23-1*28 = 23+150+23-28 = 173+23-28 =
196-28 = 168
Answer: 168'''

Your solution: '''23+25*6+23-1*28 = 23+150+23-28 = 170+23-28 = 193-28 =
165
Answer: 165'''

Correct solution: '''Let's think step by step:
23+25*6+23-1*28 = 23+150+23-28 = 173+23-28 = 196-28 = 168
Answer: 168'''
---

Question: '''What is the result of 2+22*4+17-4*24?'''

Agent 1's solution: '''2+22*4+17-4*24 = 2+80+17-96 = 82+17-96 = 99-96 = 3
Answer: 3'''

Your solution: '''2+22*4+17-4*24 = 2+88+17-96 = 90+17-96 = 105-96 = 9
Answer: 9'''

Correct solution: '''Let's think step by step:
2+22*4+17-4*24 = 2+88+17-96 = 90+17-96 = 107-96 = 11
Answer: 11'''
---

Question: '''{QUESTION_PLACEHOLDER}'''

Agent 1's solution: '''{OTHER_SOLUTION_PLACEHOLDER_0}'''

Your solution: '''{MY_SOLUTION_PLACEHOLDER}'''

Correct solution: '''Let's think step by step:
```

