# OpenReview forum: "Let Models Speak Ciphers: Multiagent Debate through Embeddings"
_ICLR.cc/2024/Conference — ICLR 2024 poster_

### Official Review · Reviewer_27LG · 2023-10-29

**Soundness:** 1 poor
**Presentation:** 2 fair
**Contribution:** 3 good
**Rating:** 5
**Confidence:** 5

**Summary:**

This paper proposes a simple but effective technique for improving the "communication" between two or more LLMs when "debating" about an answer (called CIPHER). Instead of the LM generating natural text with some token sampling technique (e.g., temperature, nucleus etc.) that is then passed to the other LM in the input context, this work generates token representations that are a weighted average of the full vocabulary. The weights are determined by the softmax predictions. So instead of natural text, the sequence of aggregated representations is passed to the other LM. The final answer is generated in natural language by falling back to the regular token sampling technique.

Authors experimented with several LLaMA models on a few reasoning tasks and found this technique to improve the final answer after a few rounds of debate between two models by 1-3.5%.

**Strengths:**

The proposed method is very simple, yet sounds interesting to explore and seems to be effective. The prompts and configurations are shared in the appendix, and the experiments are with open-sourced models so it should be easy to reproduce and build on this research.

**Weaknesses:**

While the idea of enriching the communication across models is exciting, I believe there are several shortcomings in the current work:

1. The method requires access to the output embeddings of the model and the full output probability, so cannot be directly applied to LLMs served via API.
2. While there method is described in several sections, the definition of the method still didn't feel precise enough. For example, I assume that at each step of the autoregressive generation the input token embedding is the previously aggregated token embedding and not a single sampled token? Or do you still sample tokens for decoding and just convert the full sequence to aggregated embeddings after?
3. There is a lot of discussion on the effect of sampling temperature, including ablation experiments etc., that I am confused about. From my understanding of the method (e.g., Equation 2), there is no temperature parameter in the method. The only effect of temp is in the final generation of the answer by one of the models. Therefore, I don not understand why are two temperatures reported, or the ablation in Figure 5.
4. The experimental results are only on 4 datasets and the improvements are relatively small, and no confidence intervals are reported.
5. I didn't see any discussion on validation set for hyper param setting, and according to the appendix it seems like different temperatures were used for the main baseline (NLD) and for CIPHER? Why is that? This raises some additional concerns about the experimental setup.
6. In addition to the point 2, the variants of the method are also not precisely described. For example, on Table 2: "This issue can be addressed by maintaining a bijection mapping between
the vocabulary embeddings of the two models." I can guess how it was implemented, but would appreciate elaboration.
7. Would be interesting to further understand the effect of passing these aggregated embeddings to the model. The communication to later generated tokens is anyway through K,V embeddings of the self-attention that have some dense representations and not limited to the token vocabulary. Some exploration on how the input embeddings impact the K,Vs could perhaps shed some light on how CIPHER modifies the communication

**Questions:**

please see questions in the weakness above

---

> ### Author Response · Authors · 2023-11-19
> **Reply to Reviewer 27LG [Part 1]**
>
> We thank the reviewer for the useful comments on our work. We address the comments  below.
>
> > The method requires access to the output embeddings of the model and the full output probability, so cannot be directly applied to LLMs served via API.
>
> Our research is focused on open-source language models due to its transparency and reproducibility. While open-source models provide insight into their internal mechanisms, APIs often obscure the distinction between the AI model and the engineering components, e.g., incorporating specialized engineering for specific canned responses.
>
>
> > While there method is described in several sections, the definition of the method still didn't feel precise enough. For example, I assume that at each step of the autoregressive generation the input token embedding is the previously aggregated token embedding and not a single sampled token? Or do you still sample tokens for decoding and just convert the full sequence to aggregated embeddings after?
>
> We are grateful to the reviewer for pointing out the need for greater clarity.
> In our approach, at each step $t$ of the autoregressive generation, we utilize an aggregation of all previous $t−1$ generated token embeddings as the input for the next token generation.
>
> We have revised Section 3.2 of our manuscript to reflect this clarification. The updated notation, now denoted as $\overline{emb}^{(1:t-1)}$, has been introduced to clearly represent the aggregated input at each step of the generation process. Please refer to the revised version for a more detailed and precise explanation.
>
> > There is a lot of discussion on the effect of sampling temperature, including ablation experiments etc., that I am confused about. From my understanding of the method (e.g., Equation 2), there is no temperature parameter in the method. The only effect of temp is in the final generation of the answer by one of the models. Therefore, I don not understand why are two temperatures reported, or the ablation in Figure 5.
>
> We apologize for any confusion caused by our initial presentation of Equation 2 and appreciate the opportunity to clarify. In Equation 2, our method generates a new embedding by calculating the weighted average of all token embeddings within the vocabulary set. Here, the “weight” for each token is determined by its respective probability, which is derived from the softmax of the logits. Similar to conventional natural language generation processes, the temperature parameter $T$ controls the smoothness of the probability distribution, thereby shaping the output embedding.
> To enhance clarity and facilitate better understanding, we have revised notations and Equation 2 in our manuscript. We decided to break it down into two separate equations as follows:
>
> $\bar{emb}^{(t)} = \sum_{i = 1}^{V} p_{vocab_i}^{(t)} \cdot emb_{vocab_i}$
>
> where
>
> $[p_{vocab_i}^{(t)}, \dots, p_{vocab_V}^{(t)}] = softmax \\{ logit(emb_{prompt}, \bar{emb}^{(1:t-1)} ) /T  \\}$
>
>
>
>
> In addition to the equation, we have revised Section 3.2 of our manuscript to clarify on the temperature:
>
> *”**Role of temperature.** The temperature $T$ in Eq.1 and Eq.3 controls the smoothness of the probability $p^{(t)}_{vocab_i}$. When $T \rightarrow 0$, both CIPHER's embedding generation and natural language generation result in greedy generation. In contrast, a large $T$ leads to a uniform averaging and sampling over the whole vocabulary set for CIPHER and natural language generation, respectively. We find that choosing proper temperatures for the debaters plays a pivotal role in the performance of CIPHER debate and natural language debate. Thus, to ensure fairness of our empirical evaluation, we utilize Bayesian optimization [1] to select the best performing temperatures for each method in our experiments in Section 4. Moreover, we conduct sensitivity analysis on the temperatures in Section 5.2.”*
>
> [1] Bayesian Optimization: Open source constrained global optimization tool for Python
>
>
> We believe these changes will effectively address the comments and improve the overall clarity of our method's description.

---

> ### Author Response · Authors · 2023-11-19
> **Reply to Reviewer 27LG [Part 2]**
>
> > The experimental results are only on 4 datasets
>
> We would like to clarify a minor discrepancy in the number of datasets mentioned. Our experiments were conducted on five datasets, including GSM8K, Formal Logic, High School Math, Professional Psychology, and Arithmetic.
>
>
> > the improvements are relatively small, and no confidence intervals are reported.
>
>
> Our approach demonstrates consistent improvements of 1-3.5% across five datasets and six different models without any further training or fine-tuning.  Thus, we believe that the improvement is significant.
>
> We ran 5 experiments for each method in Table 1 to obtain the confidence intervals for LLaMA2, reported as below:
>
> | Method                   | GSM8K | H.S. Math | Psychology | Formal Logic | Arithmetic |
> |--------------------------|-------|-----------|------------|--------------|------------|
> | Single Answer            | 59.5±2.0  | 38.1±2.6      | 71.5±1.1       | 46.0±2.9         | 75.0±1.7       |
> | Major@5   | 65.5±1.1  | 41.5±1.5      | **74.0±1.0**   | 44.4±2.3         | 77.6±1.4       |
> | NLD    | 66.5±0.8  | 39.4±0.9      | 73.0±1.5      | 49.2±0.9         | 83.0±1.6       |
> | CIPHER (ours)            | **67.5±0** | **41.5±0**  | **74.0±0**   | **52.4±0**     | **86.5±0**   |
>
> Overall, *Single Answer* has the highest variance, ranging from 1 to 3% across five datasets. This increased variance is attributed to its token sampling process during token generation. *Major@5* and *NLD* methods reduce variance by ensembling diverse responses. However, they still rely on a similar token sampling process, leading to relatively high variance. In contrast, *CIPHER* is not affected by this issue, thanks to its deterministic embedding generation. For any given input, *CIPHER* consistently produces the same embeddings, which are then translated into identical texts via a nearest neighbor search across the vocabulary set.
>
> It's important to note that running these baseline methods are time-consuming, regarding that we need at least 5 runs for each to obtain a valid standard error estimation. While we are working on confidence intervals for additional experiments, as of now, we can only provide these intervals for LLaMA2 in Table 1 as part of our rebuttal. We plan to include all the results in the final camera-ready version of our work.
>
>
>
> > I didn't see any discussion on validation set for hyper param setting
>
> As our approach focuses solely on inference without involving any training phase, the conventional use of a validation set for hyperparameter tuning was not applicable. Additionally, we swept over temperatures to guarantee we report the best of each method, hence no need to set a validation set.
>
>
> >  according to the appendix it seems like different temperatures were used for the main baseline (NLD) and for CIPHER? Why is that? This raises some additional concerns about the experimental setup.
>
> We employ Bayesian optimization to sweep over some temperature pairs for all the baselines and our method. For a fair comparison, we reported the best performance of each method in our paper. We also included an ablation study on the temperature, as reported in Figure 5, to give a deeper understanding of how the temperature affects debate performance.
>
> As each method prefers different temperatures to obtain the best results, it is not fair to use the same temperature. For example, Majority Voting (major@5) usually requires higher temperatures compared to *Single Answer*, to have a more diverse response for the ensemble.
>
>  That said, we still conducted some results on the same pairs of temperatures (in brackets) for both CIPHER and NLD baseline as follows:
>
> | Method                   | GSM8K (0.1, 0.5) | H.S. Math (0.1, 0.2)| Psychology (0.3, 0.8) | Formal Logic (0.3, 0.5) | Arithmetic (0.2, 0.3)|
> |--------------------------|-------|-----------|------------|--------------|------------|
> | NLD    | **65.5**  | 38.2      | 68.5      | 39.7      | **74.5**       |
> | CIPHER (ours)            | 65.0 | **40.0**  | **72.5**  | **44.4**    | 74.0   |
>
>
> In general, CIPHER works better than NLD for these temperatures reported above, but again, this is not a good setting as each method favors different temperature settings. As discussed in Sec 5.2, CIPHER often prefers low and high-temperature agents to pair together. This explains its lower accuracy compared to NLD on two datasets, as shown in the table above, particularly in Arithmetic, where CIPHER prefers a large difference in temperatures.

---

> > ### Author Response · Authors · 2023-11-19
> > **Reply to Reviewer 27LG [Part 3]**
> >
> > > In addition to the point 2, the variants of the method are also not precisely described. For example, on Table 2: "This issue can be addressed by maintaining a bijection mapping between the vocabulary embeddings of the two models." I can guess how it was implemented, but would appreciate elaboration.
> >
> > Thanks for bringing this to our attention! In Section 4.2, we’ve added more implementation details for CIPHER debates between LLaMA-65B and LLaMA2-70B as follows:
> >
> > *”While the tokenizer is shared between LLaMA-65B and LLaMA2-70B, they use distinct embeddings for each vocabulary. To tackle this issue, we keep a mapping vocab -> [$embed_{LLaMA1}$, $embed_{LLaMA2}$] for each vocabulary and compute the weighted average of the embeddings using the embeddings of the receiver. For example, to pass the message to LLaMA2-70B, we average over embed_llama2 during the CIPHER response generation from LLaMA1-65B debater. This guarantees that the output of LLaMA1-65B is encoded within the LLaMA2-70B’s token embedding space.”*
> >
> >
> >
> >
> > > Would be interesting to further understand the effect of passing these aggregated
> > embeddings to the model. The communication to later generated tokens is anyway through K,V embeddings of the self-attention that have some dense representations and not limited to the token vocabulary. Some exploration on how the input embeddings impact the K,Vs could perhaps shed some light on how CIPHER modifies the communication
> >
> > We thank the reviewer for the suggestion.
> > We have included an analysis of attention heatmaps for the human language debate baseline (NLD) and CIPHER in Appendix B3 of the updated manuscript. The heatmap figures can be found in the latest version of the paper.
> >
> > *“Fig 9 shows a comparison of attention heatmaps for CIPHER and NLD at the 45$^{th}$ decoder layer of LLaMA2-70B. These heatmaps correspond to the arithmetic question we used in Figure 1 during the last debate round. Specifically, we compute the similarity between the $q$ vector of the last token of the first agent and the $k$ vectors of its preceding 100 tokens. We can observe that NLD's heatmap exhibits uniform attention distribution, lacking intense focus on any particular segment. Conversely, CIPHER's heatmap shows some distinct bright spots, particularly around the 40$^{th}$ attention head and the 74$^{th}$ time steps. This suggests that the model's attention is highly focused on those areas, potentially indicating areas of higher relevance for the task.”*

---

> ### Comment · Reviewer_27LG · 2023-11-22
>
> Thank you for your detailed reply. The updated paper now indeed better explains the role of the temperatures in the method and clarifies some of my initial confusion. I appreciate your work on revising the paper and answering my questions.
>
> I would be very interested in examining the experimental performance of the proposed method. However, I still have two concerns with the current version and response:
> 1. I think you confused my question about confidence intervals with computing variance. CI help provide a statistically more reliable estimate of the performance. One option to obtain such is to use bootstrapping which doesn't require any new runs of the model and works for both deterministic or random outputs.
> 2. I disagree with your answer that a validation set is not required here. Sweeping hyper parameters over the test set and reporting best performance is not a valid evaluation setup. Especially given the sensitivity of the performance to the temperature value as you yourself state above and as shown in the temperature heat map figure.

---

> ### Author Response · Authors · 2023-11-23
> **Thank you for your response! [part 1]**
>
> > I think you confused my question about confidence intervals with computing variance. CI help provide a statistically more reliable estimate of the performance. One option to obtain such is to use bootstrapping which doesn't require any new runs of the model and works for both deterministic or random outputs.
>
>
>
>
> We appreciate your suggestion regarding the use of bootstrapping to estimate confidence intervals (CIs). However, while confidence intervals are valuable in many statistical analyses, it is not standard in the related literature. ​​ For example, related works such as [1, 2, 3, 4, 5, 6]  presented their findings without reporting confidence intervals. In addition to the std reported for LLaMA2-70B, we’re working on adding std for the rest of the table, and will incorporate the results in our camera ready version. While we believe that including standard deviation adequately demonstrates the robustness of our method, especially in comparison with baseline models, we still show the bootstrapped 95% confidence intervals (in brackets) for our results as follows.
>
>
>
> ---
> **LLaMA2-70 in Table 1**
>
>
>
>
> | Method                   | GSM8K | H.S. Math | Psychology | Formal Logic | Arithmetic |
> |--------------------------|-------|-----------|------------|--------------|------------|
> | Single Answer            | 59.3 (52.0, 65.5)  | 38.3 (32.6, 44.1)     | 71.5 (67.8, 75.0)      | 46.0 (36.5, 54.0)         | 75.0 (69.0, 81.0)       |
> | Major@5   | 65.7 (59.0, 72.0)  | 41.3 (35.6, 47.0)      | **74.0** (70.4, 77.4)  | 44.4 (34.9, 52.4)        | 77.6 (71.5, 83.0)      |
> | NLD    | 66.5 (59.5, 72.5)  | 39.4 (32.6, 44.0)     | 73.0 (69.4, 76.4)     | 49.2 (40.5, 57.1)         | 83.0 (78.5, 89.0)      |
> | CIPHER (ours)            | **67.5** (61.0, 74.0) | **41.5** (35.6, 47.0)  | **74.0** (70.4, 77.4)   | **52.4** (42.9, 61.0)     | **86.5** (81.5, 91.0)   |
>
>
>
>
>
>
>
> ---
> **LLaMA-65B, Table 1**
>
>
>
>
> |                   | GSM8K            | H.S. math        | Psychology       | Formal Logic     | Arithmetic       |
> |-------------------|------------------|------------------|------------------|------------------|------------------|
> | Single Answer     | 50.5 (43.5, 57.5)| 33.3 (27.8, 38.5)| 66.5 (62.7, 70.2)| 43.5 (34.9, 51.6)| 29.8 (23.5, 36.0)|
> | Major@5           | 57.8 (50.5, 64.0)| 36.7 (31.1, 42.6)| 67.0 (63.0, 70.5)| 46.8 (37.3, 54.8)| 31.0 (25.0, 37.5)|
> | NLD               | 55.5 (48.5, 62.5)| 36.7 (31.1, 42.6)| 68.5 (64.6, 71.8)| 46.0 (36.5, 54.0)| 35.0 (28.5, 41.5)|
> | CIPHER            | **58.8** (51.5, 65.5)| **38.5** (32.6, 44.4)| **70.5** (66.6, 73.8)| **50.8** (42.1, 59.5)| **36.5** (30.0, 43.5)|
>
>
>
>
>
>
>
>
>
>
>
>
>
>
>
>
>
>
> ---
>
> **Fig 3 for multiagent debates across different models**
>
>
> |            | MPT-30B         | Falcon-40B-Instruct | LLaMA-65B        | LLaMA2-Chat-70B  | LLaMA2-70B        | WizardMath-70B   |
> |------------|-----------------|----------------------|------------------|-------------------|-------------------|------------------|
> | Single Answer     | 21.0 (15.5, 27.0) | 31.5 (25.1, 38.0)    | 50.5 (43.5, 57.5)| 54.5 (47.5, 61.5) | 59.2 (52.0, 65.5) | 81.0 (75.0, 86.0)|
> | NLD     | 25.5 (20.0, 32.0) | 36.5 (30.0, 43.5)    | 55.5 (48.5, 62.5)| 61.5 (54.5, 68.0) | 66.5 (60.0, 73.0) | 83.0 (77.0, 87.5)|
> | CIPHER     | **26.0** (20.5, 32.5) | **38.0** (31.5, 45.0)    | **58.8** (51.5, 65.5)| **63.5** (56.5, 70.0) | **67.5** (61.0, 74.0) | **84.5** (79.0, 89.0)|
>
> [1] Self-Consistency Improves Chain of Thought Reasoning in Language Models
>
> [2] SELF-REFINE: Iterative Refinement with Self-Feedback
>
> [3]  Improving Language Model Negotiation with Self-Play and In-Context Learning from AI Feedback
>
> [4] Teaching large language models to self-debug
>
> [5] Encouraging Divergent Thinking in Large Language Models through Multi-Agent Debate
>
> [6] Improving Factuality and Reasoning in Language Models through Multiagent Debate

---

> > ### Author Response · Authors · 2023-11-23
> > **Thank you for your response! [part 2]**
> >
> > > I disagree with your answer that a validation set is not required here. Sweeping hyper parameters over the test set and reporting best performance is not a valid evaluation setup. Especially given the sensitivity of the performance to the temperature value as you yourself state above and as shown in the temperature heat map figure.
> >
> >
> > We agree with the reviewer that the determining hyperparameters via a validation set provides more generalizability and forms a more rigorous setup. However, we would like to emphasize that having a validation set is not a common setting in this line of work, such as [1, 2, 3, 4, 5, 6]. In addition, we note that the effect of temperature selection is ablated away in our current experiment setups. The current results do show that CIPHER, when its parameters are tuned correctly, has a higher performance compared to NLD under an optimized generalization parameter setting. From the set of questions analyzed, our preliminary findings indicate that CIPHER exhibits a higher potential than NLD, particularly after we conducted an ablation study on the temperature settings.
> >
> >
> >
> >
> > We believe that testing CIPHER under the validation setup as described by the reviewer will also likely work as we did not overfit the model onto any dataset. Meanwhile, we conjecture that if there were any hyperparameter overfitting to our evaluation dataset, it very likely happened to all of the methods, not just CIPHER. Ideally, the above arguments are best supported with experiment results.  However, given the limited time remaining in the rebuttal period, we regret we are unable to provide such results by the deadline. Nevertheless, we will work diligently to provide the requested numbers and make sure they appear on the final version of the paper should it be accepted. Specifically, we will use the current set of problems as a validation set, and conduct experiments on another set of problems using the parameters determined from the current results if applicable. We would then report the numbers for both CIPHER and NLD on both sets.
> >
> >
> >
> >
> > [1] Self-Consistency Improves Chain of Thought Reasoning in Language Models
> >
> >
> > [2] SELF-REFINE: Iterative Refinement with Self-Feedback
> >
> >
> > [3]  Improving Language Model Negotiation with Self-Play and In-Context Learning from AI Feedback
> >
> >
> > [4] Teaching large language models to self-debug
> >
> >
> > [5] Encouraging Divergent Thinking in Large Language Models through Multi-Agent Debate
> >
> >
> > [6] Improving Factuality and Reasoning in Language Models through Multiagent Debate

---

> > > ### Comment · Reviewer_27LG · 2023-12-03
> > >
> > > Thank you for your response and for reporting CI.
> > >
> > > I have decided to increase the score from 3 to 5 (marginally below the acceptance threshold) since I do find the idea here interesting and the added explanation to the temperature use to resolve my initial concern regarding that.
> > >
> > > That said, I cannot support the acceptance of this paper due to issue of missing validation set and running a bayesian hyper parameter search over the test set.
> > >
> > > Given that the paper is entirely empirical, I would expect compelling empirical evidence to support the benefits of the proposed method. Due to the missing test/ validation set, I don't find the empirical results here complete or convincing. To respond to your claim that a validation set is not need here: In some limited settings like when the temperature and other hyper parameters are being set in advance and not searched over, like in some of the referenced papers in your response, it might be acceptable to not have a separate validation set. However, here a large sweep is done over temperature values. You cannot claim that the temperature values are crucial to get right and that a bayesian optimization is needed to select them, and at the same time claim that a validation set in not needed for properly evaluating the method. I believe that not running an extensive hyper parameter search over the test set is a basic requirement for ML papers and therefore unfortunately cannot support the acceptance of the current version.
> > >
> > > A more minor suggestion for strengthening the paper is also to evaluate on more tasks.

---

### Official Review · Reviewer_LfJQ · 2023-10-30

**Soundness:** 3 good
**Presentation:** 3 good
**Contribution:** 2 fair
**Rating:** 6
**Confidence:** 3

**Summary:**

The paper is about a new way of communication among large language models (LLMs) that use embeddings instead of natural language.

The paper claims that this method, called CIPHER, can improve the reasoning ability of LLMs by avoiding information loss and encoding a broader spectrum of information. The paper evaluates CIPHER on five reasoning tasks and shows that it outperforms the state-of-the-art natural language debate methods. The paper also conducts an ablation study to explain why CIPHER works better.

**Strengths:**

* The paper proposes a novel communication protocol for large language models (LLMs) that use embeddings instead of natural language.
* The paper provides a clear and detailed description of the CIPHER method and its implementation.
* The paper also conducts extensive experiments on five reasoning tasks and compares CIPHER with the state-of-the-art natural language debate methods. The paper shows that CIPHER outperforms the baselines by a large margin on all tasks.
* The paper also performs an ablation study to analyze the impact of different components and parameters of CIPHER.

**Weaknesses:**

See Questions

**Questions:**

1. The authors conducted experiments on five common reasoning datasets, can this method be tested on agent-related leaderboards
2. In formula 2, whether the response embedding will be adjusted, how are the results of different weights?
3. Why are the results in table1 and table2 completely different, how many rounds are used in table1？
4. Can this method be used for different models with the same tokenizer, for NLD, different models can communicate each other.

---

> ### Author Response · Authors · 2023-11-19
> **Reply to Reviewer LfJQ**
>
> > The authors conducted experiments on five common reasoning datasets, can this method be tested on agent-related leaderboards
>
> We kindly ask the reviewer to clarify which agent-related leaderboards might be relevant. If the leaderboard suits our framework, we would be happy to test and report it in our final version.
>
>
>
> > In formula 2, whether the response embedding will be adjusted, how are the results of different weights?
>
> In Equation 2, our method generates a new embedding by calculating the weighted average of all token embeddings within the vocabulary set. Here, the “weight” for each token is determined by its respective probability, which is derived from the softmax of the logits. Similar to conventional natural language generation processes, the temperature parameter $T$ controls the smoothness of the probability distribution, thereby shaping the output embedding. We also added a discussion on the temperature in the revised version in Sec 3.2, as follows:
>
> *”**Role of temperature.** The temperature $T$ in Eq.1 and Eq.3 controls the smoothness of the probability $p^{(t)}_{vocab_i}$. When $T \rightarrow 0$, both CIPHER's embedding generation and natural language generation result in greedy generation. In contrast, a large $T$ leads to a uniform averaging and sampling over the whole vocabulary set for CIPHER and natural language generation, respectively. We find that choosing proper temperatures for the debaters plays a pivotal role in the performance of CIPHER debate and natural language debate. Thus, to ensure fairness of our empirical evaluation, we utilize Bayesian optimization [1] to select the best performing temperatures for each method in our experiments in Section 4. Moreover, we conduct sensitivity analysis on the temperatures in Section 5.2.”*
>
>
>
> To enhance clarity and facilitate better understanding, we have revised the notations and Equation 2 in our manuscript. We decided to break it down into two separate parts, now Eq.2 and Eq.3, as follows:
>
>
> $\bar{emb}^{(t)} = \sum_{i = 1}^{V} p_{vocab_i}^{(t)} \cdot emb_{vocab_i}$
>
> where
>
> $[p_{vocab_i}^{(t)}, \dots, p_{vocab_V}^{(t)}] = softmax \\{ logit(emb_{prompt}, \bar{emb}^{(1:t-1)} ) /T  \\}$
>
>
>
>
> > Why are the results in table1 and table2 completely different, how many rounds are used in table1？
>
> The settings in Table 1 and Table 2 are different and are stated in the captions. Specifically, in Table 1, we facilitate debates between two identical LLaMA-family models, with temperature being the only difference between the two debaters. In this case, we report the accuracies of the final responses from the debater operating at the lower temperature. In Table 2, we facilitate debates between one LLaMA-65B (first version) and one LLaMA2-70B. In this case, we report the accuracies of the final round responses from the two debaters separately for a thorough analysis.
> In both Table 1 and Table 2, the total number of rounds is 3.
> We added this to the caption for Table 1 in our revised version.
>
>
>
>
> > Can this method be used for different models with the same tokenizer, for NLD, different models can communicate each other.
>
> Yes. In Table 2, we applied CIPHER debate between LLaMA-65B and LLaMA2-70B.

---

### Official Review · Reviewer_b3L6 · 2023-11-01

**Soundness:** 3 good
**Presentation:** 3 good
**Contribution:** 3 good
**Rating:** 6
**Confidence:** 4

**Summary:**

The paper introduced a communication regime named CIPHER to allow LLMs to communicate through embedding vectors instead of natural language tokens. The authors argue that this method preserves more information and avoids information loss due to token sampling. They conducted experiments on five reasoning tasks and showed that CIPHER debate outperforms natural language debate by 1-3.5%.

**Strengths:**

1. A good idea to directly use embedding vectors to communicate between LLMs.

2. The paper provides a rigorous and comprehensive evaluation of CIPHER on five diverse reasoning datasets across multiple domains. The result showed that CIPHER consistently outperforms natural language debate.

3. The paper also conducts ablation studies and sensitivity analysis to investigate the mechanisms and factors that contribute to the performance of CIPHER.

**Weaknesses:**

1. Limited Generalizability. As the authors described in the limitations, this method is only applicable to LLMs that share a common vocabulary. For different types of LLMs, aligning embeddings is a difficult task.

2. From Figure 10, the language of CIPHER is still difficult to analyze.

**Questions:**

1. Which experiment result can support the statement "our approach can generalize across a wide array of LLMs, enabling even smaller LLMs to unlock the benefits of debate and achieve better performance than majority voting"? Is there any experiment of smaller LLMs like LLaMA-2 13B or others?

2. Why is the performance of CIPHER worse than natural language debate when Round=1?

---

> ### Author Response · Authors · 2023-11-19
> **Reply to Reviewer b3L6**
>
> > Limited Generalizability. As the authors described in the limitations, this method is only applicable to LLMs that share a common vocabulary. For different types of LLMs, aligning embeddings is a difficult task.
>
> As stated in the Limitation section, requiring a shared tokenizer is a limitation of our work and is out of the scope of this paper. Extending CIPHER debate to models with different tokenizations will require tokenization/embedding alignment, which would be an interesting direction for us to further explore. Such an alignment can potentially be tackled by utilizing methods in multimodality literature, such as [1, 2]. This literature deals with broader questions related to aligning various modalities.
>
> We would like to note that, even though CIPHER debate can only be applied to models sharing the same tokenizer, we verify that it is effective in pushing the reasoning performance of the open-source models beyond the current state-of-the-art methods.
>
> [1] BLIP-2: Bootstrapping Language-Image Pre-training with Frozen Image Encoders and Large Language Models
>
> [2] MiniGPT-4: Enhancing Vision-Language Understanding with Advanced Large Language Models
>
>
>
> > From Figure 10, the language of CIPHER is still difficult to analyze.
>
> As described in the Introduction Section: “First, we do not necessarily need to understand the intermediate debates amongst LLMs. Second, natural language generation uses only one token to represent the model’s belief over the entire vocabulary, which risks losing information embedded within the model output logits.”
> The main takeaway of our approach is to trade interpretability for information preservation during the debates. Yet, we can still interpret the communications made by the debater operating at low temperatures (e.g., blue debater, now Figure 11 in the revised version) while using another debater with a higher temperature to gather more information. In practice, keeping at least one debater at a low temperature is necessary to collect the final result of the debate.
>
>
> > Which experiment result can support the statement “our approach can generalize across a wide array of LLMs, enabling even smaller LLMs to unlock the benefits of debate and achieve better performance than majority voting"? Is there any experiment of smaller LLMs like LLaMA-2 13B or others?
>
> Prior work in LLM debate and self-critique (e.g., [1, 2]) show that only some state-of-the-art LLMs, such as GPT-3.5, GPT4 are capable of taking advantage of debates and discussion. These models are over 100B parameters and closed-source.  In our work, we propose CIPHER that unleashes the potential of smaller open-source models in debates. By “generalize across a wide array of LLMs”, we mean that one can apply CIPHER to smaller sized models such as LLaMA-family ones, ranging from 7B (reported in Appendix B.2, Fig. 8a) to 70B (table 1). We chose to mainly use LLaMA2-70B in our experiments because it’s one of the largest and most up-to-date models at that time. Additionally, we showed the results across different model architectures in Fig 3 to evaluate the robustness and generalizability of our method.
>
> [1] Self-refine: Iterative refinement with self-feedback
>
> [2] Improving language model negotiation with self-play and in-context learning from AI feedback.
>
>
> > Why is the performance of CIPHER worse than natural language debate when Round=1?
>
> In Table 2, when round=1, it means the model only gives a direct response. By choosing responses of the larger model (“LLaMA2-70B” columns) as the final answers, our method is not worse, but rather slightly better than the baseline at round=1. That said, CIPHER is designed for communication in debate settings. Thus, we need debate rounds to fully take advantage of the method. We added the results by round for ablation study purposes, to see how the performance changes over rounds.

---

> > ### Comment · Reviewer_b3L6 · 2023-11-22
> >
> > Thank you for your response; my concerns have been largely addressed.
> > Regarding Question 2, I suggest incorporating the meaning of Round=1 into the figure caption.
> > Additionally, in Figure 4(a), the performance of CIPHER at rounds = 1 seems weaker than NLD.
> > Moreover, why does the performance of LLaMA2-70B (CIPHER) decrease on Arithmetic when Round=2?

---

> ### Author Response · Authors · 2023-11-22
> **Thank you for your response!**
>
> > I suggest incorporating the meaning of Round=1 into the figure caption
>
> Thank you for the suggestion. We will incorporate the meaning of round=1 to the figure caption in the camera-ready version.
>
> > in Figure 4(a), the performance of CIPHER at rounds = 1 seems weaker than NLD
>
> In Figure 4(a), CIPHER's performance is lower than NLD in the first round. In general, the performances of NLD and CIPHER are quite similar in the first round, as shown in the performance of LLaMA2-70B and LLaMA-65B in Table 2. As discussed earlier, the first-round result is the direct response without any debate. CIPHER is designed for communication, thus we need a few rounds of debate to fully take advantage of it.
>
>
> > why does the performance of LLaMA2-70B (CIPHER) decrease on Arithmetic when Round=2?
>
> It's a good observation!  In Table 2, we paired LLaMA2-70B with a smaller model, LLaMA-65B.
>
> In the first round, LLaMA-65B's performance was low, resulting in a large gap between the two models, especially on the Arithmetic dataset.
>
> In the second round, LLaMA2-70B received responses from LLaMA-65B, which were much worse compared to its own performance. This confused LLaMA2-70B, leading to a decrease in its performance in this round. However, for LLaMA-65B, this round saw a significant boost in performance, as it obtained much better answers from LLaMA2-70B.
>
> In round 3, we observed an increase in the performance of both models after the gap between them was reduced.
>
> In general, after a decrease in the second round, LLaMA2-70B recovered and boosted its performance in the third round. Note that this drop in the second round was also the case for NLD, where LLaMA2-70B experienced a decrease in the second round.
>
> Regarding the drop in performance, an interesting question arises: how can a bad debater, such as a dummy debater, affect the final debate performance? We investigate this in Appendix B.2, where we analyze the performance upper bound (debate with an expert debater) and the lower bound when using nonsensical feedback from other debaters. We observe that debate can be detrimental when the model has low capacity (Fig. 8a), but it does not pose much harm in the case of a more capable model (Fig. 8b)

---

### Official Review · Reviewer_gCSm · 2023-11-02

**Soundness:** 2 fair
**Presentation:** 2 fair
**Contribution:** 3 good
**Rating:** 5
**Confidence:** 4

**Summary:**

The authors introduce a modified multiagent debate technique. Instead of one model's output tokens being input to the other model, the distribution over tokens is used to compute a weighted average over all token embeddings, resulting in a new embedding vector which can be directly input to the second model, bypassing its token-embedding layer. They show that this method improves upon the naive token-based debate approach by between 0.5-3.5% on various benchmarks.

**Strengths:**

Allowing networks to communicate with each by sharing token-embeddings rather than raw tokens is an interesting idea, allowing for higher-bandwidth information transmission. This method shows performance improvements shown on GSM8k, MMLU, and Arithmetic benchmarks over the more direct debate method of Du et al.

**Weaknesses:**

Although the high level ideas of the paper are interesting and potentially performance-boosting, the lack of detailed explanations and unusual formatting and presentation makes it hard to understand exactly what the authors are doing, and whether the performance improvements are actually due to their vector-sharing approach or something else.

Various technical explanations were unclear or lacking, in particular those having to do with temperature-selection:
* It is unclear how the Convert-and-Aggregate function works, in particular how the responses from multiple debaters are distilled into a single response.
* The "Result collection" and "Metrics" paragraphs in Section 4.1 are the first time in the paper that differing temperatures are mentioned. If an optimization procedure is being used for temperature selection as part of the method, then this should be described in detail along with the rest of the method in Section 3.
* The temperatures used should all be clearly reported, and whatever process is used for temperature selection should either also be applied to the other baseline methods where relevant, or ablated away in a separate experiment to highlight potential sensitivities to this hyperparameter, or both.
* In Section 4.1 you say “we select the response from **the debater operating at the lowest temperature** as the final answer”. But in Section 5.2 you say “To determine the optimal temperature pairs, we utilize Bayesian optimization (Nogueira, 2014–), and report the accuracies based on the final round response generated by **the first debater**”. These appear to be contradictory.
* In the caption for Figure 5 you say “best performance is achieved when temperature 1 is lower than temperature 2” but this is not at all apparent from these plots. The only clear take-away from them is that accuracy is high when temperature 1 is low.

In many places in the paper the notation and formatting are confusing or nonstandard, making it difficult to read:
* Using “l” to refer to a token index rather than a layer index
* Using long names and blocky Courier-esque fonts for variables (e.g. "$\texttt{embresponse}$”)
* Using the direct-sum symbol for concatenation
* Captions for Table 1 & Table 2 are above their respective figures rather than below. These tables, captions, and their adjacent paragraphs are also extremely close together.
* The micro-parentheticals in Table 2 make the overall table hard to read without adding much, I would recommend removing these or adding them as supplemental information in the appendix.
* The heatmap plot in Figure 5 is very hard to interpret. Especially on the right side of the plot, the points are very sparse, leading to artifact-heavy interpolation. I recommend coming up with a different way of presenting this information.

**Questions:**

The most unclear parts of the paper were related to the use of temperature, and the selection procedure for temperature. These should be described explicitly and clearly along with the rest of the method, rather than being scattered across the Results and Analysis sections.

The paper is more difficult to read than it needs to be due to poor notation and formatting, which should be updated to match the style guide where appropriate.

See Weaknesses section above for more specific suggestions.

---

> ### Author Response · Authors · 2023-11-19
> **Reply to Reviewer gCSm [Part 1]**
>
> We thank the reviewer for the valuable and detailed feedback. We appreciate the advice on the presentation and have revised the paper accordingly in our new version. We address the comments as below.
>
> > It is unclear how the Convert-and-Aggregate function works, in particular how the responses from multiple debaters are distilled into a single response.
>
> We write the *Convert-and-Aggregate* function in the debate algorithm within a general framework, to accommodate the various distillation strategies in the current literature. In our revision, we made extra elaboration on what such a function can be in the context of LLM debates and also justify our choice. In general, we can apply random pick, majority voting [1], involving a judge as suggested in a concurrent work [2], and lower temperature (our method). We now clarify in Section 3.2 the following:
>
> *“To close the debate, at Convert-and-Aggregate step (Line 6), we convert the embedding responses back to natural language using nearest neighbor search over the vocabulary set, then aggregate them to obtain the final response. In most cases, LLM debaters typically reach a consensus answer by the final round, as observed in [3]. When divergence in final responses occurs, majority voting [1] or random tie-breaking are often used. However, majority voting may not be suitable for open-ended questions (e.g., summarization tasks) where multiple correct answers exist, as in the game of 24 [4], and scenarios where debates involving only two agents. Thus, in our experiments, we select the response from the debater operating at the lowest temperature as the final answer. This approach is computationally efficient as it only requires running inference on one model at the final round. Meanwhile, it also achieves comparable accuracy with the best performing debater, as evidenced in Fig.5”*
>
> [1] Self-Consistency Improves Chain of Thought Reasoning in Language Models
>
> [2] Encouraging Divergent Thinking in Large Language Models through Multi-Agent Debate
>
> [3] Improving Factuality and Reasoning in Language Models through Multiagent Debate
>
> [4] Tree of thoughts: Deliberate problem solving with large language models
>
>
>
> > The "Result collection" and "Metrics" paragraphs in Section 4.1 are the first time in the paper that differing temperatures are mentioned. If an optimization procedure is being used for temperature selection as part of the method, then this should be described in detail along with the rest of the method in Section 3.
>
> Thanks for pointing it out. We have incorporated the *Result collection* section along with the temperature selection procedure to Section 3.2, answered in the above question.
> We also dedicated a section for the role of temperature and how it can affect the debate resulting in the new revision, Section 3.2 as follows:
>
> *”**Role of temperature.** The temperature $T$ in Eq.1 and Eq.3 controls the smoothness of the probability $p^{(t)}_{vocab_i}$. When $T \rightarrow 0$, both CIPHER's embedding generation and natural language generation result in greedy generation. In contrast, a large $T$ leads to a uniform averaging and sampling over the whole vocabulary set for CIPHER and natural language generation, respectively. We find that choosing proper temperatures for the debaters plays a pivotal role in the performance of CIPHER debate and natural language debate. Thus, to ensure fairness of our empirical evaluation, we utilize Bayesian optimization [1] to select the best performing temperatures for each method in our experiments in Section 4. Moreover, we conduct sensitivity analysis on the temperatures in Section 5.2.”*
>
> [1] Bayesian Optimization: Open source constrained global optimization tool for Python
>
>
>
> > The temperatures used should all be clearly reported, and whatever process is used for temperature selection should either also be applied to the other baseline methods where relevant, or ablated away in a separate experiment to highlight potential sensitivities to this hyperparameter, or both.
>
> We apologize for the confusion. The temperatures used for the reported accuracies are already included in Appendix D (tables 3, 4, 5). To further aid reproducibility, we added all other temperature settings in two additional tables (Table 6 and Table 7) in Appendix D.1. These new tables correspond to the temperatures depicted in Figure 4.
> We included a reference to this Appendix in the main text of our revised version, at the beginning of Section 4.1 “Experimental Setup”, for easier reference as follows:
>
> *“For all the methods, we use Bayesian optimization to select temperatures, which are reported in Appendix D for reproducibility of our results.”*
>
> For both CIPHER and the baselines, we swept over temperatures using Bayesian hyperparameter optimization to make sure that the accuracy reported is optimized for each method. Additionally, we also investigate the effect of temperature, as shown in Contour plots in Fig. 5 for an ablation study.

---

> > ### Author Response · Authors · 2023-11-19
> > **Reply to Reviewer gCSm [Part 2]**
> >
> > > In Section 4.1 you say “we select the response from the debater operating at the lowest temperature as the final answer”. But in Section 5.2 you say “To determine the optimal temperature pairs, we utilize Bayesian optimization (Nogueira, 2014–), and report the accuracies based on the final round response generated by the first debater”. These appear to be contradictory.
> >
> > Thank you for bringing this to our attention. When facilitating debates between identical models operating at different temperatures, we always put the lower temperature debater in the first place. So taking the first debater’s final round response is equivalent to taking the debater operating at the lowest temperature. However, we agree that the inconsistency in our presentation can bring confusion. In the revised version, we have unified the explanations of how our final responses are obtained.
> >
> >
> >
> >
> > > In the caption for Figure 5 you say “best performance is achieved when temperature 1 is lower than temperature 2” but this is not at all apparent from these plots. The only clear take-away from them is that accuracy is high when temperature 1 is low.
> >
> > We have re-plotted Fig 5 with different temperature ranges, and changed the color scale to make the accuracies more distinguishable in the figures. The main takeaway from these plots is that CIPHER debate (Fig. 5, bottom row) usually prefers the debaters to have their temperatures wider spread apart in order to achieve optimal performance. Additionally, the optimal regions appear on the left side of the charts, where temperature 1 is less than temperature 2. These indicate that a good strategy is using various temperature agents, and choosing the response of the lower temperature agent as the final debate answer.
> >
> > We updated the discussion in our new version and rewrote Sec 5.2 as follows:
> >
> > *”[...] We investigate the advantages of allowing certain debaters to deviate from natural language during debates. More concretely, one debater operates at a lower temperature to ensure the final answer remains comprehensible to humans, while the other debater is tasked with conveying information that may deviate from natural language by generating its responses at a higher temperature. We employed Bayesian optimization [1] to identify promising pairs of temperature settings. We evaluate the effectiveness of these settings based on the accuracy of the final response from the first debater, referred to as temperature 1. This approach enabled us to examine the relationship between temperature settings and debate performance, allowing us to determine whether lower or higher temperature settings yield better results”*
> >
> > *”[...] Notably, the optimal regions for CIPHER are mostly on the left side of the charts, indicating that CIPHER benefits the most when it pairs a low-temperature agent with a high-temperature one. At higher temperatures, the probability distribution of the tokens in the vocabulary becomes more uniform, allowing CIPHER's responses to lean towards less confident token choices. This effectively complements the information communicated by other CIPHER debaters operating at lower temperatures, where they focus on more confident tokens. Additionally, low-temperature agents are necessary for collecting results that can be easily understood by humans. Therefore, a good strategy for CIPHER is to employ various temperature agents and use the response of the
> > lower temperature agent as the final debate answer.”*

---

> > > ### Author Response · Authors · 2023-11-19
> > > **Reply to Reviewer gCSm [Part 3]**
> > >
> > > > In many places in the paper the notation and formatting are confusing or nonstandard, making it difficult to read:
> > >
> > > We updated all the changes as suggested as follows
> > >
> > >
> > > > Using “l” to refer to a token index rather than a layer index
> > >
> > > We switched to $t$ for token indexing.
> > >
> > >
> > > > Using long names and blocky Courier-esque fonts for variables (e.g. "embresponse”)
> > >
> > > We have revised the namings and fonts in equations and algorithms.
> > >
> > >
> > > > Using the direct-sum symbol for concatenation
> > >
> > > We changed the direct-sums to “concat”  to make it clear.
> > >
> > >
> > > > Captions for Table 1 & Table 2 are above their respective figures rather than below.
> > >
> > > Putting captions above the tables is actually required by the ICLR formatting guidelines.
> > >
> > > > These tables, captions, and their adjacent paragraphs are also extremely close together.
> > >
> > > We adjusted the paragraphs to make it clear to read.
> > >
> > > > The micro-parentheticals in Table 2 make the overall table hard to read without adding much, I would recommend removing these or adding them as supplemental information in the appendix.
> > >
> > > We have removed them to make the table clear
> > >
> > >
> > > > The heatmap plot in Figure 5 is very hard to interpret. Especially on the right side of the plot, the points are very sparse, leading to artifact-heavy interpolation. I recommend coming up with a different way of presenting this information.
> > >
> > > During our experiments, we explored the temperature pairs using Bayesian optimization, which naturally explores more in the high accuracy area, resulting in most of the points being gathered around the left side of the charts. It would be better to have more points on the right side of the plot, but due to the time constraint of this rebuttal phase, we were not able to have more experiments. Thus, we rescaled the heatmaps by exponential coloring and chose more meaningful axis ranges. We hope these changes make the plots easier to read.
> > >
> > >
> > > > The most unclear parts of the paper were related to the use of temperature, and the selection procedure for temperature. These should be described explicitly and clearly along with the rest of the method, rather than being scattered across the Results and Analysis sections.
> > >
> > > Thank you for bringing this to our attention. In accordance with this, we’ve moved the Result Collection part to Section 3.
> > > We also dedicated a section for the role of temperature and how it can affect the debate resulting in the new revision, Section 3.2 as follows:
> > >
> > > *”**Role of temperature.** The temperature $T$ in Eq.1 and Eq.3 controls the smoothness of the probability $p^{(t)}_{vocab_i}$. When $T \rightarrow 0$, both CIPHER's embedding generation and natural language generation result in greedy generation. In contrast, a large $T$ leads to a uniform averaging and sampling over the whole vocabulary set for CIPHER and natural language generation, respectively. We find that choosing proper temperatures for the debaters plays a pivotal role in the performance of CIPHER debate and natural language debate. Thus, to ensure fairness of our empirical evaluation, we utilize Bayesian optimization [1] to select the best performing temperatures for each method in our experiments in Section 4. Moreover, we conduct sensitivity analysis on the temperatures in Section 5.2.”*

---

### Author Response · Authors · 2023-11-19
**Thank you for your feedback!**

We express our gratitude to the reviewers for their insightful and helpful remarks. Their feedback, highlighting both the merits and areas for enhancement in our work, is greatly valued. Following their guidance, we have carefully revised our paper.  The revisions are highlighted in blue in the updated pdf. We are confident that these revisions have notably enhanced the overall quality and clarity. For detailed responses to each reviewer's comments, please refer to the accompanying notes below.

---

### Author Response · Authors · 2023-11-21
**Rebuttal follow-up**

Dear reviewers,

As we are approaching the end of the discussion period, we would like to see if any areas still need our attention. We are more than willing to delve deeper into any issues that you feel haven't been fully resolved yet. We believe that a more interactive dialogue could be highly beneficial in addressing any remaining concerns or areas of improvement.

We highly value your constructive feedback, which has been instrumental in refining our manuscript. Your participation in this discussion would be greatly valued.

Thank you for your time and consideration!

We look forward to hearing from you.

Best regards,

---

### Meta-Review · Area_Chair_7Tud · 2023-12-13

**Metareview:**

Multiagent debate amongst LLMs is a nascent field with many open questions: what is a good debate protocol, under what conditions is consensus achieved, and how robust and effective can such approaches be for various domains. This paper studies the difference between exchanging information via tokens or via soft embeddings, and argues through experiments that the latter is more effective through performance improvements shown on GSM8k, MMLU, and Arithmetic benchmarks.  The main weaknesses raised were: poor notation and formatting, lack of clarity on the use of temperature, limited interpretability of how the LLMs are "talking", limited generalizability in the sense that participating LLMs need to share a vocabulary, and less-than-convincing demonstrations involving smallish improvements without confidence intervals.

**Justification For Why Not Higher Score:**

The presentation aspects of the paper could be improved;  some limitations around interpretability and generalizability need further exploration. The proposed methods in their current form show smallish improvements - but they do provide evidence that debate protocols can be significantly improved.

**Justification For Why Not Lower Score:**

No questions were raised about novelty: the paper makes a valuable contribution on the design space of debating frameworks. That the method requires access to embeddings and wont work with only API access was brought up as a weakness: but in the AC's view, exploring API-only approaches is highly limiting.

---

### Decision · Program_Chairs · 2024-01-16

Accept (poster)